# Multiplexed nanomaterial-assisted laser desorption/ionization for pan-cancer diagnosis and classification

Hua Zhang [1,15], Lin Zhao[2,15], Jingjing Jiang[2,15], Jie Zheng[3,15], Li Yang[1], Yanyan Li[1], Jian Zhou [4], Tianshu Liu[5], Jianmin Xu[6], Wenhui Lou[6], Weige Yang[6], Lijie Tan[7], Weiren Liu[4], Yiyi Yu[5], Meiling Ji[6], Yaolin Xu [6], Yan Lu[2], Xiaomu Li[2], Zhen Liu[8], Rong Tian[8], Cheng Hu[1], Shumang Zhang[1], Qinsheng Hu[9], Yangdong Deng[10], Hao Ying [11], Sheng Zhong[3], Xingdong Zhang[1], Yunbing Wang [1✉], Hua Wang [12✉], Jingwei Bai[8✉], Xiaoying Li[2✉] & Xiangfeng Duan [13,14]

As cancer is increasingly considered a metabolic disorder, it is postulated that serum metabolite profiling can be a viable approach for detecting the presence of cancer. By multiplexing mass spectrometry fingerprints from two independent nanostructured matrixes through machine learning for highly sensitive detection and high throughput analysis, we report a laser desorption/ionization (LDI) mass spectrometry-based liquid biopsy for pan-cancer screening and classification. The <u>M</u>ultiplexed <u>N</u>anomaterial-<u>A</u>ssisted <u>L</u>DI for <u>C</u>ancer <u>I</u>dentification (MNALCI) is applied in 1,183 individuals that include 233 healthy controls and 950 patients with liver, lung, pancreatic, colorectal, gastric, thyroid cancers from two independent cohorts. MNALCI demonstrates 93% sensitivity at 91% specificity for distinguishing cancers from healthy controls in the internal validation cohort, and 84% sensitivity at 84% specificity in the external validation cohort, with up to eight metabolite biomarkers identified. In addition, across those six different cancers, the overall accuracy for identifying the tumor tissue of origin is 92% in the internal validation cohort and 85% in the external validation cohort. The excellent accuracy and minimum sample consumption make the high throughput assay a promising solution for non-invasive cancer diagnosis.

[1] National Engineering Research Center for Biomaterials, Sichuan University, Chengdu 610064, China. [2] Department of Endocrinology and Metabolism, Fudan Institute of Metabolic Diseases, Zhongshan Hospital, Fudan University, Shanghai 200032, China. [3] State Key Laboratory of Information Engineering in Surveying, Mapping and Remote Sensing, Wuhan University, Wuhan 430079, China. [4] Department of Liver Surgery and Transplantation, Liver Cancer Institute, Zhongshan Hospital, Fudan University, Shanghai 200032, China. [5] Department of Oncology, Zhongshan Hospital, Fudan University, Shanghai 200032, China. [6] Department of General Surgery, Zhongshan Hospital, Fudan University, Shanghai 200032, China. [7] Department of Thoracic Surgery, Zhongshan Hospital, Fudan University, Shanghai 200032, China. [8] School of Pharmaceutical Sciences, Tsinghua University, 100084 Beijing, China. [9] Department of Orthopaedic Surgery, National Clinical Research Center for Geriatrics, West China Hospital, Sichuan University, Chengdu 610041, China. [10] School of Software, Tsinghua University, 100084 Beijing, China. [11] CAS Key Laboratory of Nutrition, Metabolism and Food safety, Shanghai Institute of Nutrition and Health, Chinese Academy of Sciences, Shanghai 200031, China. [12] Department of Oncology, the First Affiliated Hospital, Institute for Liver Diseases of Anhui Medical University, Hefei 230032, China. [13] Department of Chemistry and Biochemistry, University of California, Los Angeles, CA 90095, USA. [14] California NanoSystems Institute, University of California, Los Angeles, CA 90095, USA. [15] These authors contributed equally: Hua Zhang, Lin Zhao, Jingjing Jiang, Jie Zheng. ✉email: yunbing.wang@scu.edu.cn; wanghua@ahmu.edu.cn; jingwbai@tsinghua.edu.cn; li.xiaoying@zs-hospital.sh.cn

Cancer is becoming a leading cause of death and a major concern of public health, accounting for roughly 15% of all-cause mortality worldwide[1]. Making an early and reliable diagnosis of cancer is essential for a favorable prognosis[2–4]. Up to date, there are only limited blood tumor biomarkers available, including alpha-fetoprotein (AFP), cancer antigen 19-9 (CA19-9), and carcinoembryonic antigen (CEA), whose sensitivities remain far from satisfactory. Non-invasive screening solutions for early diagnosis are indispensably needed. Recently, blood-based liquid biopsy has shown great promise as a non-invasive and sensitive approach in detecting and localizing cancers at a relatively early stage[5–7]. To this end, pan-cancer screening has attracted increasing interest with many novel technologies explored. For example, tumor-educated blood platelet (TEP) has been used for differentiating six common tumors by RNA-seq of TEPs isolated from whole blood[8]. A multi-analyte plasma test called CancerSEEK relies on cell-free DNA and circulating proteins to detect eight common cancers[9]. Surface proteins from serum extracellular vesicles have also been utilized for classifying six different cancers[10]. More recently, an approach called "DNA evaluation of fragments for early interception" was applied to diagnose and classify up to seven different cancers by detecting abnormal fragmentation patterns of cell-free DNA[11]. However, most of these approaches require complex processing of samples, sophisticated machinery, large blood sample volumes, and the accuracy remains to be improved, limiting their clinical applications.

Previous studies have demonstrated potential application of metabolomics for certain cancer diagnosis as cancer is recognized as a metabolic disorder[12]. Nevertheless, a pan-cancer-screening strategy based on metabolomics has not been developed to the best of our knowledge. Small-molecule metabolites constitute a unique source of cancer-specific information, as metabolic alterations can participate directly in the process of transformation or support the biological processes that enable tumor growth[13,14]. Utilization of metabolomics for clinical cancer research has become an area with fast-growing interest[15,16]. Mass spectrometry-based approaches are playing an increasingly dominant role in the past two decades, with tremendous growth in instrumentation, experimental protocols, and the relevant algorithms[17–20]. Both non-targeted qualitative and targeted quantitative approaches using mass spectrometry have been developed and refined to fit certain clinical or research needs. Current cancer metabolomics largely depend on a targeted strategy where molecules that can be detected and recognized are limited. In contrast, non-targeted LDI mass spectrometry has several intrinsic advantages, including prompt analysis, high-throughput and low sample consumption. However, LDI typically requires an assisting organic matrix to transfer energy to the analytes, compromising the accuracy of small-molecule metabolites detection because of the background ions from the matrix[21]. A nanomaterial-assisted approach could be well suited for studying low mass range metabolomics with LDI where these nanomaterials are capable of absorbing laser energy without generating complex cluster ions that complicate detected signals[22]. Nanomaterials have been widely used in biomedical imaging, drug delivery and cancer treatment[23–26]. Inorganic materials assisted ionization and desorption for mass spectroscopy has been reported on various nanomaterials such as noble metal nanostructures, metal oxide nanoparticles, silicon and carbon nanomaterials, etc[22,27–31]. Without the matrix ions interference, this strategy typically has higher sensitivity at low m/z region compared to traditional matrix assisted LDI, making it possible to quantitively analyze the small molecular metabolites in serum samples. However, the studies to date generally relies on a single nanomaterial matrix for LDI and analysis with limited fingerprint information, which is prone to false alarm.

Here we report a Multiplexed Nanomaterial-Assisted LDI for Cancer Identification (MNALCI) approach for pan-cancer diagnostics. By using two independent nanostructured materials as the unique matrix materials for highly sensitive multiplexed detection and combining with machine learning for high-throughput analysis, the MNALCI allow highly sensitive capture and analysis of the signals below 1000 Da for small-molecule metabolites. Since the ionization/desorption efficiency can vary among different inorganic materials and result in different sensitivity across the m/z spectrum, the MNALCI produces a multiplexed fingerprinting information of metabolites from serum samples for high fidelity cancer identification. The potential of MNALCI was demonstrated by its excellent performance in diagnosing and classifying up to six different types of common cancers using minimum serum samples, including liver cancer, lung cancer, pancreatic cancer, colorectal cancer, gastric cancer and thyroid cancer. This approach established a low-cost, high-throughput procedure based on trace amount of serum to diagnose and classify different types of cancers with high precision, demonstrating substantial potential for standard clinical practice in cancer diagnostics and beyond.

## Results

**Gold nanoshell (GNS) and porous silicon nanowires (SiNW) for LDI.** The first type of nanomaterials utilized was $Au/SiO_2$ core/shell nanoparticles. GNS was previously reported to have superior performance over other Au nanostructures such as nanospheres and nanorods in LDI application, which could be attributed to its rough surface and strong surface plasmonic effect[32,33]. In our case, Au shell was grown on the silica core to form core-shell nanospheres by absorbing pre-synthesized Au nanoseed on the silica beads and in-situ growing into continuous nanoshell[34]. The morphology and optical properties of the core/shell nanoparticles were characterized by TEM, SEM and UV–vis absorption. As shown in Fig. 1a, the nanoparticles were sized about 150 nm and exhibited relative rough surfaces, as compared to the smooth appearance of bare silica nanobeads, which was a result of nucleation and growth from multiple Au nanoparticle seeds. The elemental mapping using TEM-EDAX technique unambiguously proved the formation of core-shell structures with enriched Si and O elements in the core and Au elements in the shell, from which the shell thickness of ~17 nm was estimated (Fig. 1b, Supplementary Fig. 1). The UV–Vis absorption peaks around 680 nm of the GNS nanoparticle also revealed the formation of Au shell on insulating core structure (Supplementary Fig. 1a). Interestingly, the nanoparticle solution also demonstrated strong absorption in UV region, which could be caused by the rough nature of the shell structure[33]. This unique property may largely benefit the energy absorption from the UV laser excitation of LDI.

The second type of nanomaterials utilized was highly porous silicon nanowires obtained from n-type silicon wafer using Ag-assisted chemical etching[25]. Porous silicon or silicon nanowires were previously used for LDI-MS in small-molecule analysis to enable hydrophobic surface for higher LDI efficiency[22,35,36]. In our case, porous feature was enabled with increased concentration of $H_2O_2$ during the Ag catalyzed solution etching, which favored the interaction of small molecular metabolites with the enlarged surface area. As shown in Fig. 1c, high density nanowire forest was readily obtained with nanowire length controlled by etching time. The transmission electron microscopy (TEM) proved the porous nature of the nanowires with irregular pores in 5–10 nm region (Fig. 1c inset, Supplementary Fig. 2c). The broad absorption in the UV region may be originated from the

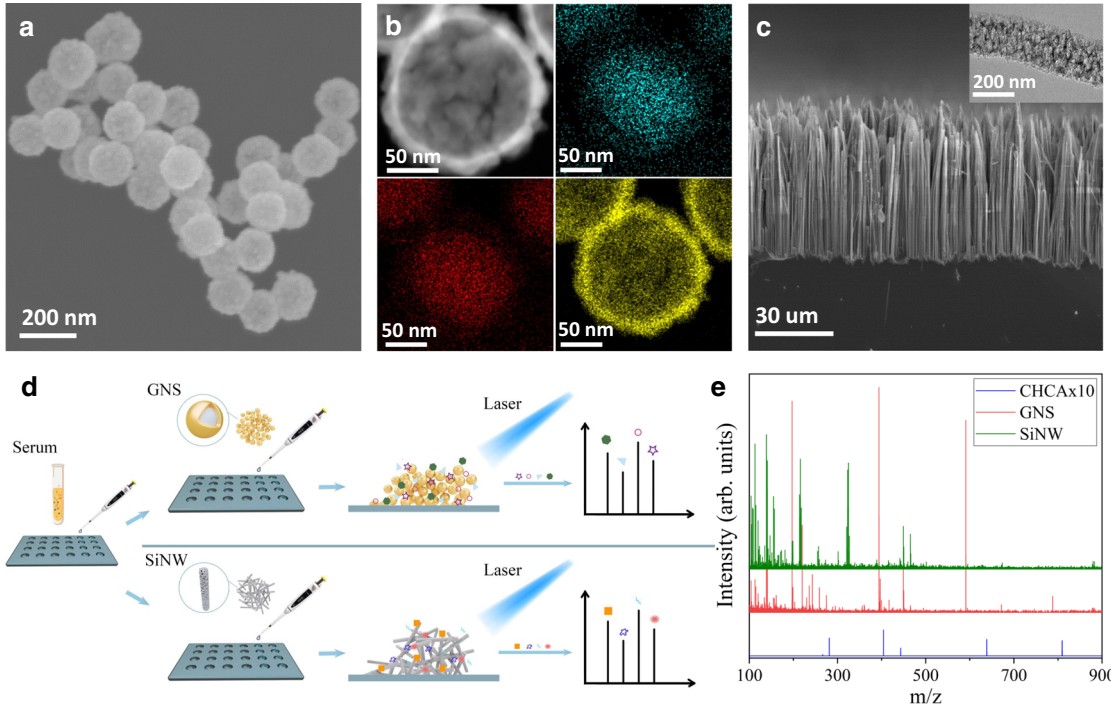

**Fig. 1 Preparation of two nanomaterials for MNALCI. a** Characterization of gold nanoshell (GNS): SEM showing relative uniform size distribution ~150 nm in diameter and the rough surface feature ($n \geq 5$ randomly selected). **b** TEM images ($n \geq 3$ randomly selected) and EDX analysis of the elemental distribution with O (azure), Si (red) and Au (yellow), proving the core-shell nanostructure. **c** Characterization of porous silicon nanowires: cross-section SEM image of the etched Si wafer showing high density nanowires forest ($n \geq 5$ randomly selected). (inset), TEM image showing highly porous nature of individual nanowires ($n \geq 3$ randomly selected). **d** Schematic of GNS and SiNW-assisted LDI measurement. **e** Comparison of typical LDI signals of a representative serum sample mixed with GNS, SiNW and CHCA.

quantum confinement in the porous nanowires with wide-range distribution of the critical dimensions. (Supplementary Fig. 2a)[37,38]. Additionally, this highly oxidative etching process generated amorphous $SiO_2$ layer covering the silicon nanocrystallined core, rendering the hydrophilic nature for water dispersion. By simply sonicating the etched silicon wafer in water, centimeter sized sample could produce milli-liter nanowire solution for thousands of samples (Supplementary Fig. 2b). This greatly reduced the material cost, making it affordable for clinical application.

In a typical procedure, 0.5 μL of serum sample from an individual was spotted on the target plate followed by 1 μL of nanomaterials and dried under room temperature where nanomaterials were utilized to assist detection of the analytes (Fig. 1d and see more details in methods). As ionization/ desorption process varied among different nanomaterials, GNS-assisted LDI generated more signal in the rage of 200–300 Da while SiNW produced more information below 200 Da. (Fig. 1e) As comparison, a direct mix with traditional α-cyano-4-hydroxycinnamic acid (CHCA) matrix and serum sample could hardly produce any signal, even at 10 times enlargement, possibly due to the interference of lipids[39]. Although lipid interference using CHCA diminished with over 50 times dilution of serum samples, this would be challenging to generate signals on low abundance metabolites, making it difficult to produce distinguishable m/z features among healthy and cancerous samples.

**Establishment of MNALCI for pan-cancer diagnosis.** Based on both GNS and SiNW-assisted LDI, MNALCI was established and tested for pan-cancer screening in two hospital-based cohorts in China. The Shanghai cohort composed of 1008 individuals that include 203 healthy controls and 805 patients diagnosed with

stage I–IV cancers according to American Joint Commission on Cancer (AJCC): liver cancer ($n = 139$), lung cancer ($n = 76$), pancreatic cancer ($n = 97$), colorectal cancer ($n = 238$), stomach cancer ($n = 119$) and thyroid cancer ($n = 136$) from Zhongshan Hospital, Fudan University in Shanghai, China. The Hefei cohort including 175 individuals that include 30 healthy controls and 145 patient with stage I–IV cancers: liver cancer ($n = 29$), lung cancer ($n = 28$), colorectal cancer ($n = 30$), stomach cancer ($n = 30$), thyroid cancer ($n = 28$) from the First Affiliated Hospital of Anhui Medical University in Hefei, China was further investigated as an external validation cohort in the present study. Unfortunately, in the Hefei cohort blood samples for pancreatic cancer were not available. The general characteristics of all the patients and controls were summarized in Table 1. These six types of cancers were selected because they were listed among top 10 cancers in China[40,41]. The diagnoses of all the cancers were made by pathological verification Specific histopathologic types of cancers were adopted in the current study, including hepatocellular carcinoma (HCC), non-small cell lung adenocarcinoma (NSCLC), pancreatic ductal adenocarcinoma (PAAD), colorectal adenocarcinoma (CRC), gastric adenocarcinoma (GC) and papillary thyroid carcinoma (PTC). None of the patients received any treatment for cancer before serum samples were collected, including surgery, chemotherapy, radiotherapy, etc. In contrast, non-cancerous healthy controls of 203 individuals had normal biochemical profiles (including serum tumor antigens), negative ultrasound (abdominal, thyroid), radiological (lung), and endoscopic (colorectum, gastric) findings, and no previous history of any type of cancer.

Both GNS and SiNW-assisted LDI tests on a serum sample yielded rich information with numerous peaks, with subtle variations among cancerous samples and controls. These multiplexed fingerprinting information by different nanomaterials were then

**Table 1 Summary of patient and healthy control clinical characteristic.**

| Patient type | Cohort | N | Gender | | Age | AJCC stage | | | |
|---|---|---|---|---|---|---|---|---|---|
| | | | M | F | | I | II | III | IV |
| HCC | Training | 111 | 93 | 18 | 55.41 ± 10.67(25–80) | 41 | 37 | 33 | – |
| | Internal | 28 | 27 | 1 | 55.54 ± 13.35(31–77) | 10 | 11 | 7 | – |
| | External | 29 | 24 | 5 | 56.66 ± 8.22(38–74) | 23 | 3 | 3 | – |
| NSCLC | Training | 60 | 30 | 30 | 57.63 ± 11.88(30–80) | 41 | 9 | 8 | 2 |
| | Shanghai | 16 | 3 | 13 | 59.50 ± 9.54(40–76) | 11 | 3 | 2 | – |
| | External | 28 | 7 | 21 | 58.46 ± 10.35(36–74) | 25 | 3 | – | – |
| PAAD | Training | 77 | 44 | 33 | 64.03 ± 8.44(47–83) | 26 | 38 | 9 | 4 |
| | Internal | 20 | 14 | 6 | 60.50 ± 9.20(45–80) | 5 | 8 | 4 | 3 |
| | External | / | / | / | / | / | / | / | / |
| CRC | Training | 191 | 119 | 72 | 60.50 ± 10.63(31–84) | 24 | 41 | 26 | 100 |
| | Internal | 47 | 32 | 15 | 58.23 ± 10.98(29–83) | 2 | 11 | 6 | 28 |
| | External | 30 | 18 | 12 | 64.57 ± 9.26(48–84) | 16 | – | 14 | – |
| GC | Training | 96 | 61 | 35 | 57.84 ± 11.24(28–81) | 2 | 8 | 33 | 53 |
| | Internal | 23 | 17 | 6 | 54.48 ± 11.77(26–76) | – | 3 | 7 | 13 |
| | External | 30 | 20 | 10 | 56.77 ± 11.60(35–84) | – | 2 | 8 | 20 |
| PTC | Training | 108 | 28 | 80 | 43.79 ± 12.13(21–67) | 94 | 13 | 1 | – |
| | Internal | 28 | 9 | 19 | 42.29 ± 9.91(27–62) | 27 | 1 | – | – |
| | External | 28 | 8 | 20 | 44.82 ± 9.30(24–62) | 26 | 2 | – | – |
| HC | Training | 163 | 93 | 70 | 47.29 ± 10.60(23–76) | / | / | / | / |
| | Internal | 40 | 24 | 16 | 49.27 ± 11.45(28–70) | / | / | / | / |
| | External | 30 | 22 | 8 | 42.40 ± 6.43(31–52) | / | / | / | / |

*HCC* hepatocellular carcinoma, *NSCLC* non-small-cell lung cancer, *PAAD* pancreatic adenocarcinoma, *CRC* colorectal carcinoma, *GC* gastric cancer, *PTC* papillary thyroid carcinoma, *HC* healthy control, *Internal* The patients and healthy controls for the internal validation set from Zhongshan Hospital, Fudan University, Shanghai, China as an internal validation cohort, *External* The patients and healthy controls for the external validation set from the first Affiliated Hospital of Anhui Medical University, Hefei, China as an external validation cohort, *M* male, *F* female, *AJCC* American Joint Committee on Cancer.

combined and analyzed by a machine learning classifier. A wrapper approach was used to conduct feature selection. An SVM recursive feature elimination (SVM-RFE) procedure was adopted to order variables to the norm of the weights. First, all data were taken to compute the norm of the weights for different features. Then the features with the smallest norm were eliminated. This process is repeated until all features are ranked[42,43]. To establish metabolic fingerprints for pan-cancer diagnosis and classification, we used an SVM based approach[44]. As a popular classification algorithm for computer vision, medical imaging, and bioinformatics, SVM finds decision boundary in a high or infinite vector space by maximizing the distance between the hyperplanes. Additionally, ensemble learning was applied to combine multiple weak classifiers into a stronger one[45].

The Shanghai cohort was split into a training cohort and an internal validation cohort (Supplementary Fig. 3). A two-step SVM procedure was devised for MNALCI to map the metabolite profiles in the training cohort, including 643 cancer patients and 163 healthy controls (Fig. 2a and Supplementary Fig. 3). The first SVM classifier separated cancerous samples and healthy controls, based on GNS, SiNW, or both. The fusion model which combined GNS and SiNW enabled a 99% sensitivity at 93% specificity as exhibited on the Receiver Operating Characteristic (ROC) curve. The sensitivity and specificity were calculated under the binary classification between cancerous samples and healthy controls. The area under the curve (AUC) of the fusion model was 0.994 for distinguishing cancers from healthy controls, which was superior to either GNS or SiNW-assisted model alone (Fig. 2b). After discriminating cancer patients from healthy controls, a second SVM classifier was used to distinguish among six different cancers. As shown in the confusion matrix, under the specificity of 93%, the overall accuracy of fusion model was 91% for multi-cancer classification where the individual accuracy varied between 83% for NSCLCs and 98% for HCCs and PAADs among different cancer types and controls (Fig. 2c). In contrast, the accuracy of GNS-assisted SVM model was 87% at 82%

specificity (Supplementary Fig. 4a), while that of SiNW-assisted SVM model was 86% at 84% specificity (Supplementary Fig. 4d).

To evaluate the accuracy of MNALCI, we first tested with the single-blinded internal validation cohort, including 162 patients and 40 healthy controls. For distinguishing cancers from healthy controls, the fusion model had 93% sensitivity at 91% specificity, and an AUC of 0.999 (Fig. 2d). At 91% specificity, the fusion model for multiclass cancer discrimination had an overall accuracy of 92%, ranging from 82% for NSCLCs to 100% for PAADs, HCCs and PTCs (Fig. 2e). In contrast, the accuracy of GNS-assisted model was 84% at 86% specificity, while that of the SiNW-assisted model was 85% at 78% specificity (Supplementary Fig. 3b, e). Then we tested with another single-blinded external validation cohort (Hefei cohort), including 145 patients and 30 healthy controls. The fusion model had 84% sensitivity at 84% specificity, and an AUC of 0.990 (Fig. 2f). At 84% specificity, the fusion model for multiclass cancer discrimination had an overall accuracy of 85% (Fig. 2g). The accuracy of GNS-assisted model fell to 77% at 73% specificity and that of SiNW-assisted model fell to 77% at 84% specificity (Supplementary Fig. 4c, f).

In a cancer-screening scenario, it is imperative to have very high specificity to avoid false positive results and unnecessary anxiety. To this end, a threshold value (θ) was introduced. As θ increases, the specificity increases while the sensitivity decreases (see supplementary for more details). For the fusion model, the highest accuracy was achieved when θ was set to 1.0 (Supplementary Data 1). When the specificity was raised to 98% as θ increases, the overall accuracy of cancer classification persisted at 87% (Supplementary Fig. 4a). This result was much better than single nanomaterial-assisted models (Supplementary Figs. 5d and 4g), which is consistent seen in both internal and external validation results (Supplementary Fig. 5).

**Clinical implications of MNALCI.** Besides cancer identification and classification, MNALCI also provided important clues for

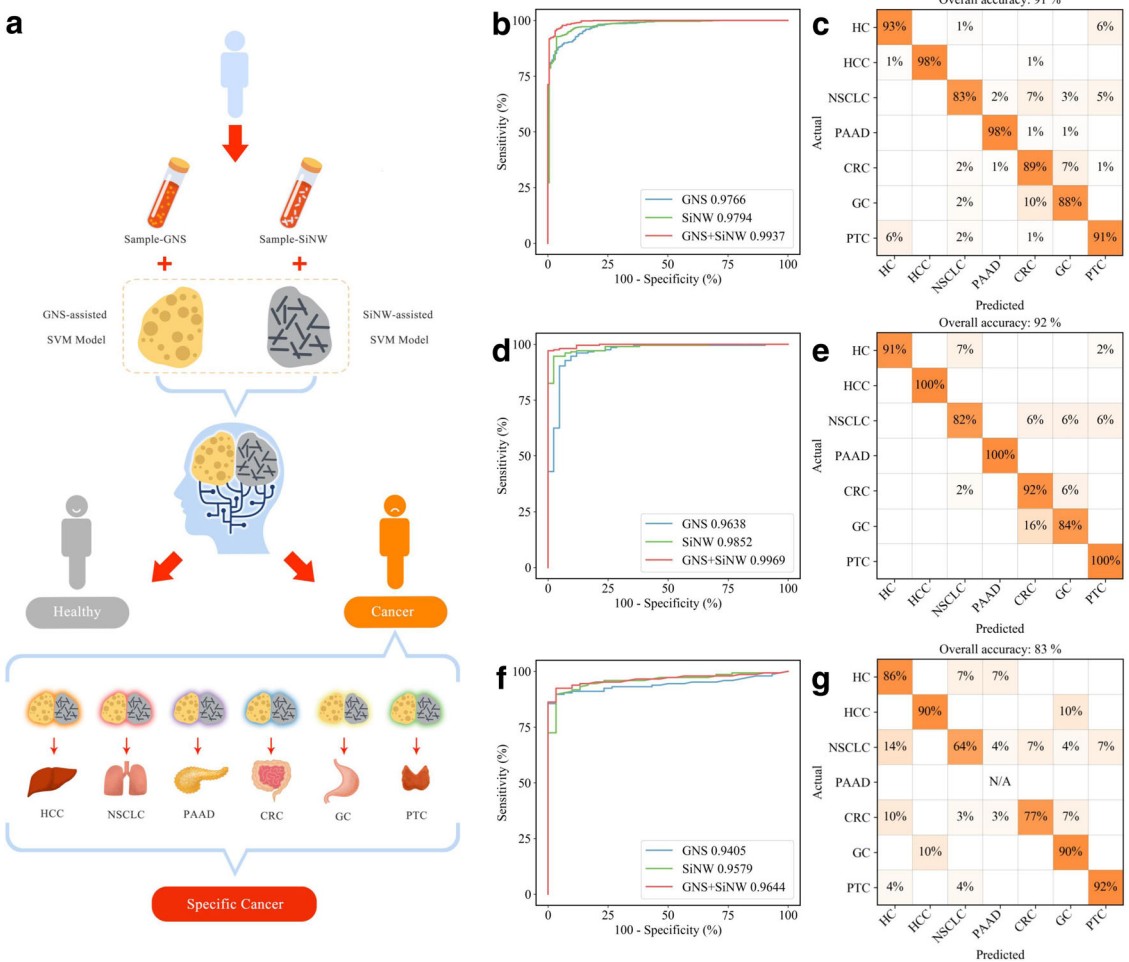

**Fig. 2 Detection and classification of cancers by MNALCI.** Flow Diagram for MNALCI (**a**). ROC curves with the best accuracy of the GNS-assisted SVM model, SiNW-assisted SVM model and the fusion model for distinguishing patients from healthy controls in the training cohort (**b**), internal validation cohort (**d**) and external validation cohort (**f**). Confusion matrix summarizing the cancer classification results in the training cohort (**c**), internal validation cohort (**e**) and external validation cohort (**g**) using the fusion model.

potential small-molecule biomarkers. After training with cancerous patients and healthy controls, top 10 discriminative m/z features of each cancer type vs healthy control along with their corresponding P values were selected for GNS and SiNW-assisted LDI, respectively (Fig. 3a–i and Supplementary Data 2). Unlike a box plot in which all of the plot components correspond to actual datapoints, the violin plot features a kernel density estimator of the underlying distribution that illustrates the overall probability density. Heatmaps with top 30 features were also generated for visualization of the differences among six types of cancers as well as healthy controls (Supplementary Figs. 6 and 7). The unique metabolites, represented by different clusters of m/z values, underscored the differences in metabolome found in the circulation of each cancer type. Furthermore, across these top m/z peaks for different cancer types, 8 metabolite biomarkers were identified and confirmed with LDI MS/MS and LC-MS/MS (Supplementary Figs. S8–S23), namely 2-oxovaleric acid, histamine, glucose, 5-hydroxymethyluracil, 2-Furoic acid, methylmalonic acid, 4-methylcatechol and L-carnitine. Most of these metabolites were reported elsewhere for the link of cancer events[46–52]. Importantly, amongst all 120 m/z features, these 8 metabolites correspond up to ~30 signals from Fig. 3, while some of the other m/z features may attribute to different ionization or fragments of the metabolites. It is also interesting to notice that some of the metabolites showed a preference for GNS or

SiNW-assisted LDI. For example, 5-hydroxymethyluracil, which corresponded to m/z = 164.98, appeared 4 times in Fig. 3 with GNS-assisted LDI while methylcatechol, which corresponded to m/z = 147.02, also appeared 4 times in Fig. 3 with SiNW-assisted LDI, indicating different interaction mechanism between analytes and nanomaterials. In addition, all LDI MS/MS spectra of the discriminatory features presented in Fig. 3 were showed in the supplementary materials (Supplementary Fig. 24–35).

The ability to detect cancers at relatively early stages is one of the most important attributes for an excellent diagnosis. In the present study, AUCs of almost all cancer types reached 1.000 in the training cohort, internal validation cohort and external validation cohort under ideal settings: each individual cancer group alone vs. healthy controls, without the interference of other cancers (Supplementary Fig. 36). Next, we further generated ROC curves for each type of cancer using all patients vs healthy controls of Shanghai cohort according to different cancer stages, with the exception of GC due to limited number of early-stage patients in our cohort. The AUC for the earliest-stage cancers (stage I) was 0.999 for HCC, 0.922 for NSCLC, 1.000 for PAAD, 0.998 for CRC and 0.969 for PTC. Actually the median AUCs were similar for all four stages, namely 0.998 for stage I cancer, 0.9992 for stage II (ranging from 0.935 for NSCLS to 1.000 for PAAD), 0.9997 for stage III (ranging from 0.966 for NSCLC to 1.000 for HCC and PAAD), and 0.9998 for stage IV (ranging

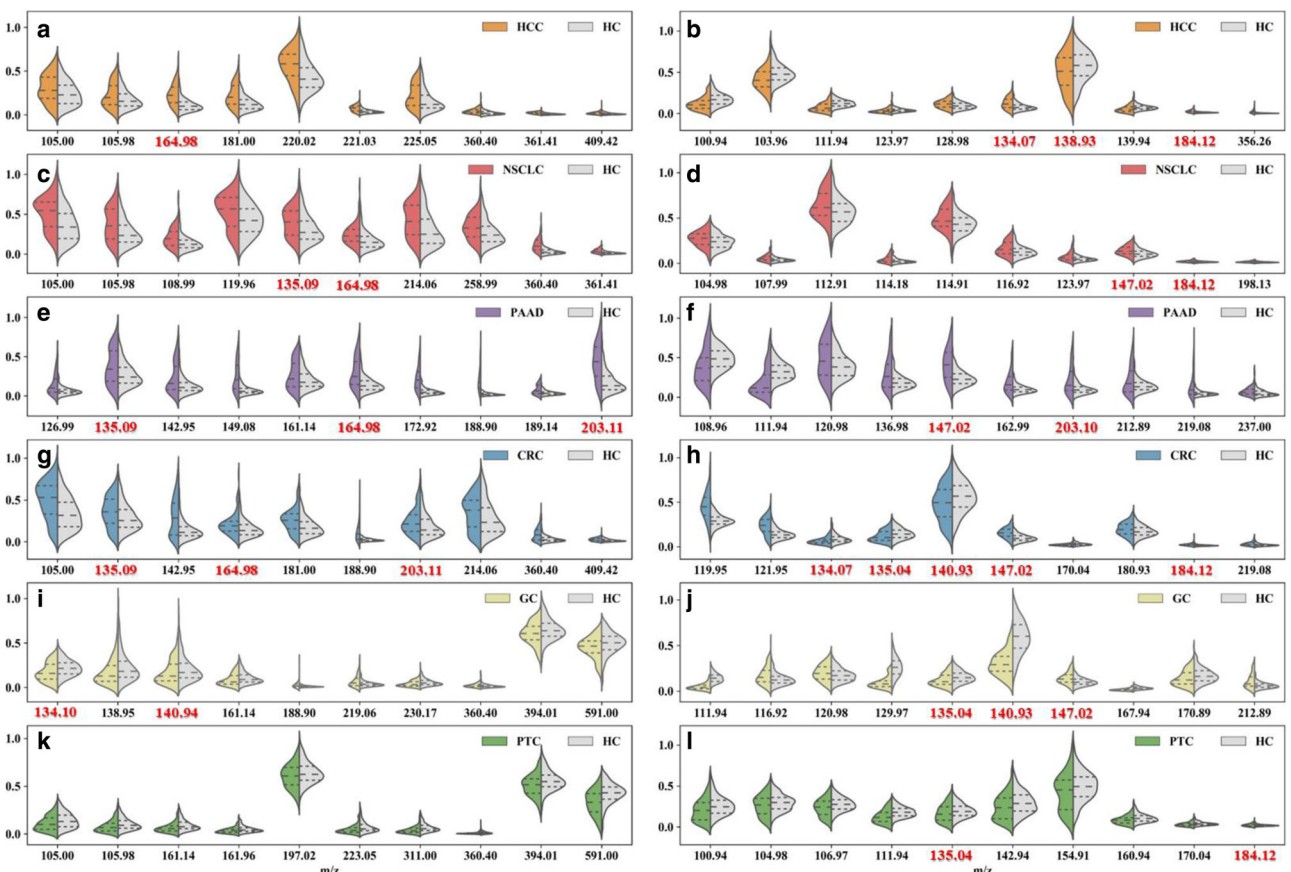

**Fig. 3 Discriminating features of each cancer type versus healthy controls.** Violin plots of top 10 m/z LDI intensity distributions of each cancer type vs healthy control chosen by MNALCI. (The middle dash lines indicated median value of the LDI intensities of each corresponding m/z while the upper and lower dotted lines indicated intensity values of first quartile and third quartile. Gray represented healthy control while the colored represented cancer patients). **a** HCCs vs healthy controls by GNS-assisted LDI. **b** HCCs vs healthy controls by SiNW- assisted LDI. **c** NSCLCs vs healthy controls of GNS-assisted LDI. **d** NSCLCs vs healthy controls by SiNW-assisted LDI. **e** PAADs vs healthy controls by GNS-assisted LDI. **f** PAADs vs healthy controls by SiNW-assisted LDI. **g** CRCs vs healthy controls by GNS-assisted LDI. **h** CRCs vs healthy controls by SiNW-assisted LDI. **i** GCs vs healthy controls by GNS-assisted LDI. **j** GCs vs healthy controls by SiNW-assisted LDI. **k** PTCs vs healthy controls by GNS-assisted LDI. **l** PTCs vs healthy controls by SiNW-assisted LDI. It should be noted that all red bolded numbers represented for certain metabolites discovered in this report. (Supplementary Figs. 8–23).

from 0.998 for PAAD to 1.000 for HCC) (Supplementary Fig. 37). These results demonstrated huge potential of MNALCI spotting patients with early-stage cancers.

Among the 6 different types of cancers included in the current study, HCC, PAAD and CRC have relatively sensitive and specific serum tumor antigen markers: AFP for HCC, CA19-9 for PAAD and CRC, and CEA for CRC, respectively. Those three serological markers are commonly used for cancer screening, as well as surveillance after surgery or other treatment. We compared MNALCI and those three tumor markers for HCC, PAAD and CRC detection. In the Shanghai cohort, 47 of 137 HCC patients (35.77%) were AFP negative (AFP < 20 ng/ml, stage I $n = 18$, stage II $n = 17$ and stage III $n = 12$); 11 of 94 PAAD patients (11.70%) were CA19-9 negative (CA19-9 < 37 U/ml, stage I $n = 2$, stage II $n = 7$, stage III $n = 1$ and stage IV $n = 1$); 109 of 232 CRC patients (46.98%) were CEA negative (CEA < 5 ng/ml, stage I $n = 24$, stage II $n = 35$, stage III $n = 22$ and stage IV $n = 28$) and 149 of 230 CRC patients (64.78%) were CA19-9 negative (CA19-9 < 37 U/ml, stage I $n = 24$, stage II $n = 47$, stage III $n = 25$ and stage IV $n = 53$). Here, our study showed that MNALCI recognized almost all AFP negative HCC, CA19-9 negative PAAD, CEA or CA19-9 negative CRC. Only 2 of 137 HCCs (1.46%), 1 of 94 PAADS (1.06%), 1 of 232 CRCs for CEA (0.43%) and 2 of 230 CRCs for CA19-9 (0.87%) were misclassified as healthy controls (Fig. 4a–d). In contrast, AFP positive and AFP

negative HCCs could not be well distinguished by this method, neither could CA19-9 positive/negative PAADs or CEA positive/ negative CRCs (Supplementary Fig. 38). These results suggested that MNALCI was independent of and superior to these tumor markers.

Hotspot gene mutation tests of tumor tissues have emerged as the basis for tumor molecular pathology, laying foundation for tumor classification and precision medicine. In the Shanghai cohort, 36 of 69 NSCLC tumors had EGFR mutation (52.17%), 49 of 103 CRC had KRAS mutation (47.57%), and 91 of 130 PTC had BRAF mutation (70%). However, MNALCI showed no significant difference between EGFR mutant patients and wild-type patients in NSCLC, KRAS mutant and wildtype patients in CRC and BRAF mutant patients and wildtype patients in PTC ($p = 0.680$, 0.198 and 0.103, respectively). (Supplementary Fig. 39). Therefore, MNALCI was unable to provide information related to cancer genetic status.

## Discussion

Unlike the widely used GC/LC-MS, MNALCI adopts biochemical signatures derived by non-targeted semiquantitative mass spectroscopy for diagnosis. Instead of relying on a single biomarker, MNALCI utilizes all metabolite signals detected between 100–1000 Da. In clinical laboratories, GC/LC-MS analysis of small metabolites is typically developed with the knowledge of

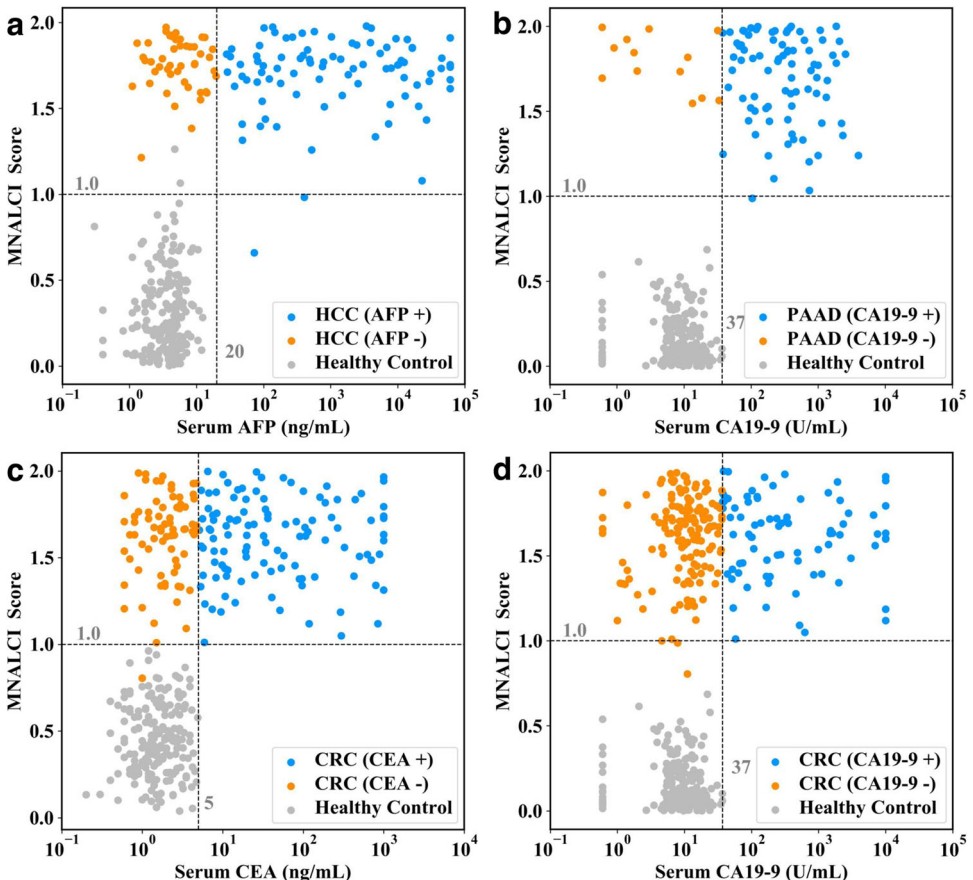

**Fig. 4 Comparison between MNALCI and serum tumor antigens in diagnosing specific cancers.** The cutoffs for MNALCI score (the horizontal line) were set to 1.0 for the highest accuracy while the cutoffs for tumor antigens (the vertical line) were as per manufacturer's recommendation. **a** Comparison of the probability by MNALCI with serum AFP for the detection of HCC and healthy control. **b** Comparison of the probability by MNALCI with serum CA19-9 for the detection of PAAD and healthy control. **c** Comparison of the probability by MNALCI with serum CEA for the detection of CRC and healthy control. **d** Comparison of the probability by MNALCI with serum CA19-9 for the detection of CRC and healthy control. Fusion models were applied to all four figures.

their chemical structures[53,54]. In contrast, the MNALCI does not require the knowledge of chemical structures of specific meta-bolites, but mainly relied on the overall fingerprint of a group of metabolites, which could be statistically quantified and compared. This strategy is ideally suited for low-cost and high-throughput diagnosis. Here, we showed that serum metabolome revealed by MNALCI could offer valuable diagnostic information for up to six different major cancers, clearly demonstrating that trace amount of serum could be employable for pan-cancer screening. The subtle alteration of metabolome underlying each type of cancer was adequately captured by this method. It would be interesting to characterize, investigate and compare the key metabolites underlying those discriminative m/z values among different cancers in future mechanistic studies.

In the present study, we took advantage of SVM and ensemble learning. SVM scaled relatively well to high-dimensional data, provided good generalization, and delivered a global solution. Thanks to two independent nanomaterials, ensemble learning combined two weak classifier (GNS and SiNW) into a strong one (fusion model), which significantly improved the overall accuracy of both cancer diagnosis and classification. For example, in the internal validation test, the fusion model avoided 20 mistakes during the binary classification process and 17 mistakes in the cancer classification process, which could have been inaccurately classified using single nanomaterial-assisted model (Supplementary Fig. 40, Table 10). GNS and SiNW varied considerably in

their ionization/desorption efficiencies, resulting in different sensitivity across the m/z spectrum for LDI, thus providing the rationale for utilizing two nanomaterials.

It should be reiterated that MNALCI does not mean to replace other methods developed for liquid biopsy. On the contrary, we believe this approach could be combined with screening strategies based on other biomarkers, such as mRNA, miRNA, mutated or 5-Hydroxymethylated cfDNA, circulating proteins, etc[5,8,55,56]. As shown in Supplementary Fig. 39, our assay was unable to dis-criminate common pathogenic mutations in NSCLC, CRC and PTC. Nevertheless, liquid biopsies based on cfDNA mutations rely heavily on the presence of driver gene mutations, whose sensitivity could be compromised for those cancers with negative driver mutations, which accounted for a significant portion of cancers[9]. The combination of multiple liquid biopsy strategies could provide additional information and further enhance the accuracy in diagnosing cancers and identifying the tumor tissue of origin.

Despite these promising results, the current study has several limitations. First, the sample size of the external validation cohort was small and without PAAD patients. Future studies should ideally include multi-center blinded cohorts with larger size and comparable composition of cancers for validation. Secondly, for each type of cancer, the total number of patients included was relatively small for machine learning. If more cancer types were included, the accuracy for classifying each individual cancer could be compromised. Nevertheless, the accuracy of MNALCI is

expected to increase if more high-quality data are available for training. Third, the discrimination relied mainly on the clusters of m/z and some of the metabolites (Fig. 3a–i and Supplementary Fig. 8–23), without knowing the identities of all the small metabolites. Finally, systemic factors such as chronic or transient inflammatory diseases, metabolic diseases, and other non-cancerous diseases may also influence the metabolic profile. These interfering conditions should be evaluated in future studies.

Our study has laid conceptual and practical foundation for a cost-effective and high-throughput technique for screening multiple cancers. The cost of the test is estimated to be less than $100 in the foreseeable future and a fully equipped lab can facilitate the measurement and analysis of over 3000 samples per day. The 6 cancer types studied here accounted for 3,929,000 newly diagnosed cancers and 2,338,000 (65%) of the estimated cancer deaths in China in 2015[41,57]. Prospective studies in a large population will be required to validate its clinical utility. With more optimization and data training, MNALCI could ultimately be translated to clinics to assist precision medicine, helping define and locate common malignancies.

## Methods

**Preparation of silicon nanowires (SiNWs).** n-Si (100) wafer with a resistivity of 0.008–0.02 Ω·cm was used for porous silicon nanowires through a two-step method: Firstly, Si pieces were immersed into a buffered oxide etchant for 2 min to remove the native oxide layer. Then, the Si pieces were immediately transferred into an Ag deposition solution containing 4.8 M HF and 0.005 M AgNO3 for 1 min at room temperature. The Ag-deposited silicon pieces were sufficiently rinsed with deionized water to remove extra silver ions and then immediately soaked into an etchant bath composed of 4.8 M HF and 0.8 M $H_2O_2$ for 60 min. At the target etching time, the Si pieces were through washing by water to remove surface Ag as well as HF residues. Afterwards, the Si pieces were immersed into concentrated HNO3 to remove Ag residues and obtain the pure silicon nanowires for 1 h. The nanowire solution was prepared by simply sonicating the etched wafer in DI water (Supplementary Fig. 2b).

## Preparation of gold nanoshells (GNS)

*APTES functionalization of silica acid.* 0.5 ml silica nanosphere (120 nm, Tianjing Daer Science and Technology Ltd, also see the SEM of Supplementary Fig. 1d) solution of concentration (2.5%) were mixed with 10 mL ethanol, and ultrasonicated for 2 min to obtain uniform dispersion. Then, 150 ul 3-aminopropyltriethoxysilane (APTES) was added into the sol-gel solution under stirring, followed by 3 h heat batch treatment at 90 °C. After cooling to room temperature, the as-functionalized nanoparticles were washed three times in ethanol by centrifugation at 2700 × g for 10 min, and finally resuspended in 2 mL ethanol.

*Modification of silica core with gold seeds.* Colloidal Au nanoseed was synthesized by Tetrakis (hydroxymethyl) phosphonium chloride (THPC) reduction of chloroauric acid. The THPC solution was prepared by mixing 0.5 mL of 1 M NaOH, 12 mL of THPC and 47.5 mL of $H_2O$. Under rapid stirring, 2.06 mL of 1 wt% aqueous chloroauric acid (HAuCl4· 3H2O) was quickly added and a color change to medium brown color can be observed in 1 min, indicating the formation of gold nanocolloids. The final solution was stored and aged at 4 °C for at least 12 h before use. The size of the seeds is estimated to be around 4 nm according to the UV absorption (Supplementary Fig. 1b). To decorate the silica nanospheres with gold nanoseeds, 2 mL amine-functionalized silica nanospheres in ethanol was mixed with 2 mL above gold seed solution and 6 mL water at ambient condition and kept for 10 min and transferred to 4 °C for at least 24 h. Finally, the gold-seeded silica nanospheres were washed three times, resuspended in 2 mL water and stored at 4 °C (Supplementary Fig. 1e).

*Growth of gold nanoshells.* 25 mg of potassium carbonate were dissolved in 100 mL DI water, and 2 mL of 1% HAuCl4· 3H2O was added to produce a colorless solution. This solution was aged at least 24 h before use. While stirring vigorously, 20 ul of gold nanoseeds modified silica nanoparticles were added to 8 mL of the above solution. After stirring for 10 min, 50 μL formaldehyde (37%) as reducing agent was slowly added, and continue to stirring for 24 h for complete reduction. The resulted production was washed with DI water for at least three times before following biosensing usage (Supplementary Fig. 1f).

**Clinical samples.** Human serum samples in the training and internal validation cohort were collected at Zhongshan Hospital Fudan University in Shanghai, China. Serum samples in the external validation cohort were collected at the First Affiliated Hospital of Anhui Medical University in Hefei, China. All cancer diagnoses were based on pathology. All the serum samples were collected before treatment,

including surgery, chemotherapy, radiotherapy, etc. All samples were anonymized, and only the gender, age and cancer-related lab results and pathological diagnosis were recorded. The healthy control serum samples in the training and internal validation cohort were collected at the Medical Examination Center (MEC) of Zhongshan Hospital Fudan University and control samples in the external validation cohort were collected at the MEC of the First Affiliated Hospital of Anhui Medical University. All healthy controls had normal biochemical profiles (including serum tumor antigens), negative ultrasound/radiological findings and no previous history of any type of cancer. All relevant ethical regulations were complied with. The study was approved by the Ethics Committee of Zhongshan Hospital Fudan University and the First Affiliated Hospital of Anhui Medical University. Written informed consent was obtained for all participants.

**LDI test of serum samples.** All MS measurements were performed on an Autoflex Max mass spectrometer (Bruker Daltonics, Bremen, Germany), within a mass range of 100–1000 Da, while the spectra were manually examined using the FlexAnalysis 3.4 software (Bruker Daltonics, Bremen, Germany). In a typical process, 0.5 uL of serum samples were spotted on a polished steel target plate MTP 384 and air-dried followed by another 1 uL of GNS or SiNW nanomaterials. The MS spectra were acquired in the reflection positive mode with smartbeam-II laser at 355 nm with a laser frequency of 1000 Hz.

A random walk of 25 shots at raster spot and 20 different spots were measured for each individual sample, therefore, 500 satisfactory shots were obtained. Evaluation parameters were set so that only spectra containing at least one peak with a resolving power of greater than 300 and a signal-to-noise ratio of more than 30 in the m/z range of 100–500 were accumulated. The MS/MS measurements were conducted under a TOF/TOF LIFT mode.

The overall performance of the mass spectrometer was checked every 9 samples as a group using a stocked human serum standard (aliquoted and frozen). In addition, a "home-made" standard consisting serine (m/z = 105.09), glucose (m/z = 180.16), tryptophan (m/z = 204.23), sucrose (m/z = 342.29), maltotriose (m/z = 504.44) and amylopentaose (m/z = 828.72) was test each run for calibration. After calibration for each run, all the standard spots were collected for a PCA analysis, the results were considered consistent when the intensity data of the standard spectrum were gathered within a predetermined range. Any outlier standard will need to be retested altogether with the other 8 samples in the same group.

**LC-MS/MS verification**

*Sample preparation.* 100 μL of 80% methanol was added into a 2 mL centrifuge tube with 50 μL of serum samples. The samples were vortexed for 10 s before ultrasonic oscillation at 4 °C for 30 min. After which, the samples stood at 4 °C for 60 min and centrifuged at 13,780 × g at 4 °C for 10 min. The supernatant was then taken and sampled for LC-MS/MS analysis.

*LC-MS/MS analysis.* Instruments:
  Liquid Chromatography: Waters ACQUITY UPLC;
  Mass Spectrometry: (AB SCIEX 5500 QQQ-MS).
  Column: Waters Acquity UPLC HSS T3 (1.8 μm, 2.1 mm × 100 mm)
  Chromatographic separation conditions: Column temperature: 40 °C; Flow rate: 0.30 mL/min; Mobile phase composition: mobile phase A is water +0.1% formic acid, mobile phase B is acetonitrile; Running time: 5 min; injection volume: 5 uL.

*MS parameters.* Ion source: ESI+; Curtain Gas: 35 psi; Collision Gas:9 arb; Ion Spray voltage: −4500 V; Temperature: 450 °C; Ion Source Gas1:55 arb; Ion Source Gas2: 55 arb.

*Multiple reaction monitoring (MRM) acquisition.* The development process of the multiple reaction monitoring (MRM) mass spectrometry method was as follows. First, the parent ion Q1 was found, then the collision energy was changed from small to large, and the parent ion was crushed. Different ion pairs are produced according to different energy sizes. During the development of the method, the most appropriate collision energy was optimized to form ion pairs with obvious characteristics and high response. The selected specific parent ions were induced to collide, and the interference of other ions was eliminated. Only the selected specific parent ions were collected by mass spectrometry. Sub ions should be at least three times stronger than the parent.

The gradient of mobile phase

| Time (min) | Flow rate (mL/min) | %A | %B | Curve |
|---|---|---|---|---|
| Initial | 0.300 | 90.0 | 10.0 | Initial |
| 1.00 | 0.300 | 90.0 | 10.0 | 6 |
| 3.00 | 0.300 | 10.0 | 90.0 | 6 |
| 4.00 | 0.300 | 10.0 | 90.0 | 6 |
| 4.10 | 0.300 | 90.0 | 10.0 | 6 |
| 5.00 | 0.300 | 90.0 | 10.0 | 6 |

**Threshold value θ**. After MNALCI scores (see Details of Data Analysis) were obtained from all the samples, a threshold value θ was introduced as a parameter that determined the boundary between cancerous patients and healthy controls, where:

$$\begin{cases} \text{cancerours, MNALCIscore} \geq \theta \\ \text{healthy, MNALCIscore} < \theta \end{cases}$$

As θ increases, more samples were considered healthy, hence the specificity increases while the sensitivity decreases. θ ranges between 0.0 and 1.0 for single model and the highest accuracy (minimum false positive and false negative) was achieved when θ was set to 0.5. For fusion model, θ ranges between 0.0 and 2.0. The highest accuracy was achieved when θ was set to 1.0 (Supplementary Data 1). However, in a large population screening scenario, we wanted to ensure high specificity to avoid misdiagnosing healthy controls. Detailed calculations of the internal and external validation cohorts were presented in Supplementary Data 3–6.

**Overall workflow of data analysis**. The overall workflow was composed of data preprocessing, model training and model validation. An internal validation cohort from Zhongshan Hospital, Fudan University and an external validation cohort from the First Affiliated Hospital of Anhui Medical University were used to test MNALCI. After the threshold value (θ) was set for different specificity, SVM model was built to compute the average validation error.

*Data preprocessing*. In order to obtain reliable results, the Mass Spectrometry dataset of Zhongshan cohort was randomly divided into two partitions. A part of 80% was used as a training cohort; the other 20% was left as internal validation cohort to verify the supervised machine learning model. The training cohort and the (internal and external) validation cohort were prepared separately but using the same strategy. First, like other analytical platforms, the raw data were preprocessed with several data preprocessing steps such as baseline correction and noise reduction. This was followed by normalization step, which could ensure reproducible comparisons. Finally, a calibration procedure was employed.

*Training model*. The supervised decision-making model proposed for cancer discrimination had following steps:

1. Five times five-fold cross-validation. We divided the training cohort into two subsets, one was T-training cohort with 80% of training cohort, the remaining 20% was a T-validation data set. This process would be repeated for 5 times to get the average training error.
2. Model training. Support Vector Machine (SVM) was applied as the classifier, data from T-training cohort was considered as input, then the trained SVM model was tested by the T-validation cohort and we could get the training error. After we developed all the 5 times 5-fold cross-validation, the average train error was obtained to train the hyper-parameters. In the end, the training model was built by the whole training cohort with the hyper-parameters.

*Validation model*.

1. Internal validation. When a threshold value was selected, the internal validation dataset was used to compute the internal validation error.
2. External validation. Another single-blinded external validation cohort from the First Affiliated Hospital of Anhui Medical University was tested to obtain the final external validation error.

**SVM**. Support vector machine (SVM), a supervised machine learning method, was usually applied to the two-class classification problem using a hyperplane. This learning strategy was to maximize the margin, and SVM solution would be transformed into a convex quadratic programming problem. It showed great advantages solving high-dimensional small sample classification problems. The basic idea of SVM classification was to find a hyperplane to divide the data, and used the support vector to maximize the segmentation of the data. The basic mathematical model was:

$$\max_{\mathbf{w},\mathbf{b}} \frac{1}{\|\mathbf{w}\|}$$
$$\text{s.t. } y_i(\mathbf{w}^T \cdot \mathbf{x_i} + b) - 1 \geq 0, \, i = 1, 2, \cdots, n$$

where $(\mathbf{x_i}, y_i)$ was the sample point of the data set, and $(\mathbf{w}, b)$ was the hyperplane.

**Ensemble learning**. Ensemble learning optimized the algorithm by building and combining multiple classifier systems. In order to customize a pan-cancer diagnostic model, an important model of ensemble learning, namely voting method was applied to provide better predictive results. For separating cancer and health

controls, majority-voting model was used:

$$H(x) = \begin{cases} c_j, & \text{if } \sum_{i=1}^{T} h_i^j(x) > 0.5 \sum_{k=1}^{N} \sum_{i=1}^{T} h_i^k(x) \\ \text{reject, otherwise} \end{cases}$$

where $h_i$ was a single classifier, $\{c_1, c_2, \cdots, c_N\}$ was the set of categories, $h_i^j(x)$ was the output of classifier $h_i$ on category tag $c_j$.

For the multiclass cancer classification, we implemented the One vs. Rest (OvR) approach: six cancer probabilities were separately obtained by the soft voting method where the highest predicted result was chosen as the output.

**Details of data analysis**. A two-step SVM approach was applied to discriminate in the validation cohort. Firstly, in the binary classification step, SVM was used to differentiate cancers and healthy controls based on GNS-assisted and SiNW-assisted model, respectively. The classification result was obtained by the fusion model when the MNALCI score (sum of the probabilities of GNS and SiNW-assisted models) was greater than 1.0. Secondly, for the multiclass cancer discrimination step, a One vs Rest (OvR) approach was constructed. After which each model of the corresponding cancer had a probability by adding the probabilities of GNS and SiNW-assisted models, and the specific cancer was indicated with maximum probability.

For the relatively complex multiclass cancer discrimination step, six examples with detailed calculation were selected to exemplify different situations (Supplementary Data 7, threshold = 1.0):

(a) Both single models classified correctly: For HCC14386 (patient ID), both GNS-assisted and SiNW-assisted models classified the patient into HCC correctly, with the highest probabilities (0.998 and 0.988, respectively) among all the six cancers. As a result, the fusion model also had the highest sum probability of 1.986 for HCC.

(b) Fusion model classified correctly but single model did not: For CRC266 (patient ID), GNS-assisted model misclassified the patient into NSCLC with probability 0.493, while both SiNW-assisted and fusion models classified the patient into CRC with the probability 0.652 and 1.105. For GC404 (patient ID), GNS-assisted model classified the patient into HCC with probability 0.904 so did fusion model (with a probability of 1.356), but the SiNW-assisted model misclassified the patient into CRC with the probability 0.546.

(c) Single model classified correctly but fusion model did not: For Lungca053 (patient ID), GNS-assisted model classified the patient into GC with a probability of 0.665, which was the correct category. The SiNW-assisted and the fusion model classified the patient into CRC incorrectly with the probability 0.928 and 0.954, respectively. For CRC206 (patient ID), SiNW-assisted model classified the patient into CRC (the correct category) with a probability of 0.569, however, both fusion model and GNS-assisted model classified the patient into CRC incorrectly with probabilities of 0.959 and 1.237.

(d) Neither single model classified correctly: For Lungca028 (patient ID), GNS-assisted model misclassified the patient into CRC with the probability 0.533 while SiNW-assisted model misclassified the patient into GC with probability 0.527, neither of which was correct. As a result, the fusion model also fell to the wrong category, with the highest sum probability 1.060 for CRC.

**Statistical analysis**. Statistical significance of the data was calculated at 95% ($p < 0.05$) confidence intervals, which were calculated by binomial distribution. MALDIquantForeign version 0.12 package in R version 3.4.4 to convert the original data from mzml format to csv format. Data processing and machine learning were carried out with the Python programming language (Python version 3.7.3), Numpy library version 1.17.2 and Pandas library version 0.25.1 for data processing and Scikit-learn version 0.21.3, Scipy version 1.3.1 for feature selection and machine learning. The confusion matrix, scatter diagram and ROC diagram are drawn by matplotlib version 3.1.1. The violin diagram is drawn by seaborn version 0.9.0. To test the accuracy of the cancer discrimination, ROC curves were applied for three different situations (GNS, SiNW, GNS + SiNW). ROC curves and AUC values were performed by Scikit-learn (https://scikit-learn.org).

**Reporting summary**. Further information on research design is available in the Nature Research Reporting Summary linked to this article.

## Data availability

The complete data used in this study are not publicly available due to ethical restrictions. The complete data that support the findings of this study (e.g., age, sex, LDI data, etc.) are available from the corresponding author for research purpose only and the request will generally be answered to within 2 weeks.

## Code availability

The custom computer code utilized in this study can be found at https://github.com/zhengjiewhu/MNALCI.

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

## Acknowledgements

Y.W. acknowledges the financial support from the Program of Introducing Talents of Discipline to Universities (111 Project, B16033). X.L. acknowledges the National Key Research

and Development Program of China (2016YFC1304801). L.Z., H.Y. and J.J. thank the National Natural Science Foundation of China (81970748, 91957205 and 82070821). J.B. acknowledges the Tsinghua University-Peking University Center for Life Sciences (CLS).

## Author contributions

H.Z., L.Z., J.J., Y.W., H.W., J.B., X.L. and X.D. designed and planned the study, and developed experimental protocols. C.H., S.Z., Q.H., Y.D., H.Y., S.Z. and X.Z. optimized the experimental protocols. H.Z., L.Z., J.J., J.Z., Z.L. and R.T. performed the experiments. L.Z., J.J., J.Z., T.L., J.X., W.L., W.Y., L.T., W.L., Y.Y., M.J., Y.X., Y.L., X.L. and H.W. organized patient enrollment, sample collection, and clinical data curation. H.Z. and J.Z. analyzed and interpreted data. H.Z., L.Z. J.J. and X.D. wrote the manuscript and incorporated feedback from all authors. H.Z., L.Z. J.J. and J.Z. contributed equally to this study.

## Competing interests

The authors declare no competing interests.
