## [Peer Review File · Nature Communications]

Reviewers' Comments:

Reviewer #1:

Remarks to the Author:

This is a review report on the manuscript entitled "Multiplexed Nanomaterial-Assisted Laser Desorption/Ionization for Pan-cancer Diagnosis and Classification", which has been submitted for potential publication in the Nature Communications journal.

In this study, Multiplexed Nanomaterial-Assisted laser desorption/ionization (LDI) mass spectrometry was used to measure small molecules in plasma. Analyses were performed on samples collected from 1,183 individuals, including 233 healthy controls and 950 patients with liver, lung, pancreatic, colorectal, gastric, thyroid cancers from two independent cohorts. Two separate nanostructured materials (Gold NanoShell, GNS and Porous Silicon Nanowires, SiNW) were used as the unique matrix materials allowing multiplexed detection of the signals of small molecules below 1,000 Da. Fingerprints obtained using two different nanomaterials were analyzed by a machine learning classifier.

The authors claim to discover a panel of signals obtained for small-molecule metabolites that can be used to diagnose six abovementioned cancers.

It is not always clear which calculations were performed on separated or combined data generated with GNS or SiNW. Although it is stated that multiplexed fingerprinting information was combined, some results are presented for separate data sets (Figures 2 and 3, Supplementary Figures S3-S6). Therefore, it is unclear whether the results presented in Figure 4 and Supplementary Figures S7 and S8 were obtained for combined data or not. It should be clearly stated in the Figures' legends.

Looking at the age of the Healthy control group, it seems to be younger than other cancer groups, except PTC. As metabolome is highly influenced by many factors, including age, sex, (Darst et al. Aging (Albany NY). 2019 Feb 28; 11(4): 1262-1282) and BMI (Cirulli et al., 2019, Cell Metabolism 29, 488-500), in a perfect situation, groups of patients being compared should be matched to these factors. No information about the participants' BMI is provided. Sex differences between HC and selected cancer groups are clearly observed.

The study group was quite large (including small validation cohort); machine learning methods were used to select significant variables. However, a detailed description of MS raw data treatment is missing, e.g., what was used to normalize the data? What was the reproducibility of the signals' measurement?

The most significant limitation of this study is the lack of identification of detected m/z signals. Without identification, the method cannot be called highly sensitive. Moreover, it is not known if some of the top signals from both methods were overlapping. In such a case, the quality of the combined models could be improved due to the addition of information about already included metabolites.

In general, the idea of using MNALCI is interesting, but without the identification of significant signals, it is impossible to evaluate the novelty of the results obtained.

Reviewer #2:

Remarks to the Author:

This work reports the use of two separate nanostructured matrices (Au/SiO₂ core/shell nanoparticles and Porous Silicon Nanowires) mixed with a tiny volume of serum for further analysis by laser desorption/ionization (LDI) mass spectrometry. This produces distinctive mass spectral profiles that are used by a support vector machine (SVM) model to discriminate cancer from healthy samples and across six different cancers.

The work is interesting and potentially very useful. Surface-based mass spectrometry is faster and cheaper than LC-MS and GC-MS, and the accuracy shown in the paper is quite promising. The strongest points are the large training cohort and the comparison between MNALCI and serum tumor antigens in diagnosing specific cancers (Figure 4).

However, the work, as currently presented, is still very preliminary and key pieces of information are missing, which hampers reproducibility and biological relevance of the work.

1. The authors use the term "metabolic fingerprints for pan-cancer diagnosis and classification", however no single metabolite is reported. This is in my opinion the weakest point of the manuscript. What is the identity of the top 30 or top 10 relevant m/z ions? MS/MS data is missing in that regard.

In order to strengthen the accuracy and robustness of this LDI approach and the SVM model, the top m/z features should be identified and validated using a complementary technique such as LC-MS. If the quantitative relationships of these top m/z features are important for the SVM model, their quantification by LC-MS should reinforce the classification algorithm and provide biological insights about these biomarkers.

In my opinion, the authors cannot sustain this method and a publication in Nature Communications on just mass spec features that could perfectly result from redundant signals from the same metabolite (isotopes, adducts, in-source fragments). In addition, the raw data preprocessing steps (code? algorithm?) are not provided for evaluation. What type of normalization process was applied? Why is Figure 3 not showing normalized values? Furthermore, HRMS data should be reported with 4 decimals.

2. SiNWs:

It is unclear how the silicon nanowire solution is obtained.

- How are the SiNWs removed from the wafer without damaging the wires (pores, length, etc)? How the authors break the wires down from the silicon wafer and how exactly are they transferred to a solution in the form of nanoparticles?
- What is the composition of the SiNWs solution? It would be helpful to show TEM images of this suspension where the wires are "floating" around. Is there any aggregation of SiNWs?
- Define "porous and 1D feature": how is the array of silicon nanowires 1D?
- Describe the "critical dimensions" of the porous nanowires: diameter, length, pore size, etc...
- Is the SiO₂ layer enough to stabilize SiNWs in solution? The authors claim that the SiO₂ layer is hydrophilic and there are pores on the wires (from the TEM in Figure 1). However, the TEM image is not very clear and should be magnified in order to demonstrate that the pores are on the wire and not on the silicon wafer surface.

3. GNS:

- Please show TEM images (as Supp Inf) from the 3 stages of the preparation: (i) silica NP; (ii) silica NP with Au seeds; and (iii) silica NP with Au shell.
- What does relative rough surface mean? What is the difference between GNS and silica NP either by TEM image or other methods (AFM?). What is the thickness/roughness of the gold shell? Authors claim that silica nanospheres are 120 nm, but at the same time the gold nanoshells are also 120 nm.
- Modification of silica core with gold seeds: Is the solution kept in the dark? Given the color change, what is the size of the Au nanoparticles? UV vis measurement or DLS (dynamic light scattering) should provide a quick evidence.
- What is the composition of the GNS solution that is deposited on the LDI target plate?

4. Why the GNS ionization is different? Which part of the GNS mostly contributes to the LDI MS ionization efficiency? Is it only the gold? Can you omit the silica nanoparticle core? How critical is the architecture of the nanomaterial for the ionization process? How is the interaction analyte-nanomaterial?

This also relates to the quantitative capabilities of this approach in serum samples (Figure 1e). What is the reproducibility of SiNWs and GNS-based mass spectra? LDI MS approaches are not particularly quantitative nor reproducible. What is the quantitative dynamic range? Some features in figure 3 show intensity distributions close to 0k. Is there any signal-to-noise (S/N) ratio threshold applied to filter ions for the SVM model? Why is Figure 3 showing absolute intensity values instead of normalized values?

Half microL of serum sample from an individual was spotted on the target plate followed by 1 uL of nanomaterials. How do you control the concentration of nanomaterials in a 1 microL solution?

Reviewer #3:

Remarks to the Author:

The authors present some impressive results using laser desorption of serum samples from nanomaterial surfaces for the diagnosis of a variety of different cancers. The sample size and independent validation at a second collection site provide some confidence that the methodology has promise for identify patterns of metabolite signals that could be useful in cancer diagnosis. However, the fact that none of the signals were identified, nor even the total number of peaks used in the high dimensional analysis is a red flag. Also the use of SVM, which generates an enormous number of higher order terms, none of which were specified in the analysis could cause problems. Given a mass spectrum containing hundreds of peaks, it's not hard to imagine that the classifier contains thousands of terms many of which are non-linear and which outnumber the number of samples. The problem is that when anything changes (and it will!) like the nano-material prep, the sample prep, the sample procurement protocol, the reaction products caused by laser irradiation of the sample, etc. it's easy for some of the non-linear terms to change dramatically, throwing off the algorithm.

It's also interesting to note that the authors don't try to use a classifier with a more limited number of peaks for the overall analysis, just heatmaps. Similarly, from the violin plots in Figure 3, there are very few strong biomarker candidates here, so that even if the authors were able to identify some of them, they would need many signals to make a suitable classifier. The field of metabolomics has long moved on from the development of profiles based on a set of completely unknown peaks, because of the potential problems noted above, as well as the need for a biological connection between the metabolic changes observed and the disease state that can be verified or validated with further experiments.

For these reasons, I do not believe that the paper should be accepted for publication in Nature Communications. If, however, the authors are able to identify at least some of the major contributors to their model (i.e., the 30 metabolite signals they show in their heatmaps), then I think the paper would potentially be publishable, based on the positive qualities mentioned above.

Minor point: I don't think that the very high AUC values match with some of the sensitivity and specificity numbers provided. For example, it is not possible to generate an AUC of 0.99 using sensitivity=0.84 and specificity 0.84. The highest AUC could be in that situation is $1-(0.16)^2 = 0.97$, but it could definitely be lower depending on the shape of the AUC curve. The authors need to double check their other AUC values as well.

Reply to Referee #1:

Comments: *This is a review report on the manuscript entitled “Multiplexed Nanomaterial-Assisted Laser Desorption/Ionization for Pan-cancer Diagnosis and Classification”, which has been submitted for potential publication in the Nature Communications journal.*

In this study, Multiplexed Nanomaterial-Assisted laser desorption/ionization (LDI) mass spectrometry was used to measure small molecules in plasma. Analyses were performed on samples collected from 1,183 individuals, including 233 healthy controls and 950 patients with liver, lung, pancreatic, colorectal, gastric, thyroid cancers from two independent cohorts. Two separate nanostructured materials (Gold NanoShell, GNS and Porous Silicon Nanowires, SiNW) were used as the unique matrix materials allowing multiplexed detection of the signals of small molecules below 1,000 Da. Fingerprints obtained using two different nanomaterials were analyzed by a machine learning classifier.

The authors claim to discover a panel of signals obtained for small-molecule metabolites that can be used to diagnose six abovementioned cancers.

Response: We thank the referee for carefully reading the manuscript, and raising a number of excellent points about our studies. We welcome the opportunity to address these questions, and describe the changes we have made accordingly in the manuscript.

Comment 1: *It is not always clear which calculations were performed on separated or combined data generated with GNS or SiNW. Although it is stated that multiplexed fingerprinting information was combined, some results are presented for separate data sets (Figures 2 and 3, Supplementary Figures S3-S6). Therefore, it is unclear whether the results presented in Figure 4 and Supplementary Figures S7 and S8 were obtained for combined data or not. It should be clearly stated in the Figures’ legends.*

Response: Thanks for bringing up this very important point. We have re-written the Figure caption to eliminate confusion. To further clarify, we have added in the manuscript that when we used the term “MNALCI”, it implied a “Multiplexed” result, i.e., a combined model.

Comment 2: *Looking at the age of the Healthy control group, it seems to be younger than other cancer groups, except PTC. As metabolome is highly influenced by many factors, including age, sex, (Darst et al. Aging (Albany NY). 2019 Feb 28; 11(4): 1262–1282) and BMI (Cirulli et al., 2019, Cell Metabolism 29, 488–500), in a perfect situation, groups of patients being compared should be matched to these factors. No information about the participants’ BMI is provided. Sex differences between HC and selected cancer groups are clearly observed.*

Response: Thanks for the very insightful question. It’s true that the healthy control group is younger than other cancer group except PTC, and the BMI of healthy control group is larger than other cancer groups except PTC. And as we know the male to female ratio in liver cancer incidence is about 2~3 and the difference is stronger in high-incidence than in low-incidence areas (Cancer Letters 286 (2009) 9–14). So there are more male HC patients than female patients. In order to further clarify the fact that age, sex or BMI were not significant parameters for MNALCI, we made the following steps:

- i. Age: We divided the healthy control group to younger group (younger than 40) and elderly group (older than 55), we used the same model and younger and elderly could not be distinguished by this method ($p = 0.492$ by t-test).

- ii. Sex: We divided the healthy control group based on sex; yet male and female showed no difference using this model ($p = 0.641$ by t-test).
- iii. BMI: We divided the healthy control group to normal BMI group (BMI less than 23) and obesity group (BMI more than 28), no difference of NALCI SCORE was observed between these two groups ($p = 0.367$ by t-test).

These additional analyses demonstrate that age, sex, and BMI do not play a significant role in the screening process.

Figure R1-1. a, Binary classification models of age (a), sex (b) and BMI where Y axis represents MNALCI output. The probability shows no significant difference between two groups in any of these three figures.

Comment 3: The study group was quite large (including small validation cohort); machine learning methods were used to select significant variables. However, a detailed description of MS raw data treatment is missing, e.g., what was used to normalize the data? What was the reproducibility of the signals' measurement?

Response: Thanks and we agree these are important details and have now included the details in the method section of the revised manuscript. Specifically, the first step after input of the raw data is smoothing by applying a square root transformation ($f(x) = \sqrt{x}$) to the Poisson distributed data to normal data. Second, we removed background effects to reduce their influence in quantification of the peak intensities. This was then followed by alignment method, which divided the metabolite signals 100-1,000 Da with a step size of 0.025, and added the metabolite signals of each interval. A fourth step of normalization was using the Total Ion Current (TIC) method to enable quantitative comparison across multiple spectra. Finally, the csv format data was obtained.

Figure R1-2. Flowchart of MS raw data pre-treatment.

To ensure reproducibility, evaluation parameters were set so that only spectra containing at least one peak with a resolving power of greater than 300 and a signal-to-noise ratio of more than 30 in the m/z range of 100-1,000 were accumulated. The overall performance of the mass spectrometer was checked every day using a “home-made” standard and a stocked human serum standard prior to each run (please see revised supplementary materials for more details).

Comment 4: *The most significant limitation of this study is the lack of identification of detected m/z signals. Without identification, the method cannot be called highly sensitive. Moreover, it is not known if some of the top signals from both methods were overlapping. In such a case, the quality of the combined models could be improved due to the addition of information about already included metabolites.*

Response: We appreciate the very insightful comment. In our revision, 8 metabolite biomarkers were identified and confirmed with MS/MS, namely 2-oxovaleric acid, histamine, glucose, 5-hydroxymethyluracil, 2-Furoic acid, methylmalonic acid, methylcatechol and L-carnitine. Most of these metabolites were reported elsewhere for the link with cancer events (see revised manuscript for details). Some of the other m/z features may be attributed to different ionization or fragments of the metabolites. For example, $m/z = 219.08$ could well be $[\text{glucose}+\text{K}]^+$ and $m/z = 105.00$ could be fragment from histamine (Supplementary Fig. S9). This way, amongst all 120 m/z features, these 8 metabolites correspond up almost a third of the signals from Figure 3. It is also interesting to notice that some of the metabolites showed a preference for GNS or SiNW-assisted LDI. For example, 5-hydroxymethyluracil, which corresponded to $m/z = 164.98$, appeared 4 times in Figure 3 with GNS-assisted LDI while methylcatechol, which corresponded to $m/z = 147.02$, also appeared 4 times in Figures 3 with SiNW-assisted LDI, indicating different interaction mechanism between analytes and nanomaterials, and such multiplex approach with different nanomaterial may help to improve the fidelity of our approach.

Reply to Referee #2:

Comments: This work reports the use of two separate nanostructured matrices (Au/SiO₂ core/shell nanoparticles and Porous Silicon Nanowires) mixed with a tiny volume of serum for further analysis by laser desorption/ionization (LDI) mass spectrometry. This produces distinctive mass spectral profiles that are used by a support vector machine (SVM) model to discriminate cancer from healthy samples and across six different cancers.

The work is interesting and potentially very useful. Surface-based mass spectrometry is faster and cheaper than LC-MS and GC-MS, and the accuracy shown in the paper is quite promising. The strongest points are the large training cohort and the comparison between MNALCI and serum tumor antigens in diagnosing specific cancers (Figure 4).

However, the work, as currently presented, is still very preliminary and key pieces of information are missing, which hampers reproducibility and biological relevance of the work.

Response: We thank the referee for carefully reading the manuscript, and raising a number of excellent points about our studies. We welcome the opportunity to address these questions, and describe the changes we have made accordingly in the manuscript.

Comment 1: The authors use the term “metabolic fingerprints for pan-cancer diagnosis and classification”, however no single metabolite is reported. This is in my opinion the weakest point of the manuscript. What is the identity of the top 30 or top 10 relevant m/z ions? MS/MS data is missing in that regard.

In order to strengthen the accuracy and robustness of this LDI approach and the SVM model, the top m/z features should be identified and validated using a complementary technique such as LC-MS. If the quantitative relationships of these top m/z features are important for the SVM model, their quantification by LC-MS should reinforce the classification algorithm and provide biological insights about these biomarkers.

In my opinion, the authors cannot sustain this method and a publication in Nature Communications on just mass spec features that could perfectly result from redundant signals from the same metabolite (isotopes, adducts, in-source fragments).

Response: We thank the reviewer for the very insightful comment. In our revision, 8 metabolite biomarkers were identified and confirmed with MS/MS, namely 2-oxovaleric acid, histamine, glucose, 5-hydroxymethyluracil, 2-Furoic acid, methylmalonic acid, methylcatechol and L-carnitine. Most of these metabolites were reported elsewhere for the link of cancer events (see revised manuscript for details). Some of the other m/z features may attribute to different ionization or fragments of the metabolites. For example, m/z = 219.08 could well be [glucose+K]⁺ and m/z = 105.00 could be fragment from histamine (Supplementary Fig. S9). This way, amongst all 120 m/z features, these 8 metabolites correspond up almost a third of the signals from Figure 3. It is also interesting to notice that some of the metabolites showed a preference for GNS or SiNW-assisted LDI. For example, 5-hydroxymethyluracil, which corresponded to m/z = 164.98, appeared 4 times in Figure 3 with GNS-assisted LDI while methylcatechol, which corresponded to m/z = 147.02, also appeared 4 times in Figures 3 with SiNW-assisted LDI, indicating different interaction mechanism between analytes and nanomaterials.

In our SVM classifier, we used embedded method for feature selection which was called support vector machine recursive feature elimination (SVM-RFE). The idea of SVM-RFE is that

if the orientation of the separating hyperplane found by the SVM is orthogonal to a particular feature dimension then the feature is informative. SVM-RFE uses the weighting power of an SVM classifier to produce a feature ranking, and then recursively eliminates the feature with the smallest weight. The classifier was then modified by the resulting accuracy instead of the quantitative data to determine which features were included. Therefore, the MNALDI score resulted from accumulated analytes with different weighing power. In our case, the quantification of analytes using LC-MS might be altered by pretreatment procedure while LDI MS/MS remained identical with the protocol of the samples.

Inspired by the reviewer, we have carefully analyzed and confirmed 8 metabolite biomarkers with LDI MS/MS with the highest weighting power in MNALDI. This work showed that a combination of the various metabolites contributed together to the “cancerous micro-environment”.

Comment 2: *In addition, the raw data preprocessing steps (code? algorithm?) are not provided for evaluation. What type of normalization process was applied? Why is Figure 3 not showing normalized values? Furthermore, HRMS data should be reported with 4 decimals.*

Response: We appreciate the opportunity to address any confusion. We provided a new version Figure 3 of normalized values. The first step after input of the raw data is smoothing by applying a square root transformation ($f(x) = \sqrt{x}$) to the Poisson distributed data to normal data. Second, we removed background effects to reduce their influence in quantification of the peak intensities. This was then followed by alignment method, which divided the metabolite signals 100-1,000 Da with a step size of 0.025, and added the metabolite signals of each interval. A fourth step of normalization was using the Total Ion Current (TIC) method to enable quantitative comparison across multiple spectra. Finally, the csv format data was obtained.

Figure R2-1. Flowchart of MS raw data pre-treatment.

Comment 3: *SiNWs:*

-It is unclear how the silicon nanowire solution is obtained.

Response: Thanks and we agree these are all very important details and have now included the details in the method section of the revised manuscript. The silicon nanowire solution is obtained by immersing the etched wafer in DI water and sonicate, so that the nanowires is removed from the wafer by ultrasonic mechanical forces and direct dispersed in the DI water to form nanowires solution.

- What is the composition of the SiNWs solution? It would be helpful to show TEM images of this suspension where the wires are "floating" around. Is there any aggregation of SiNWs?

Response: Optical Microscopy and SEM images of SiNWs deposition on silicon substrate from the NW suspensions shows that the individual NWs can be separated, with some wires bundles which may be attributed to incompleting etching, and particles due to sonication.

Figure R2-2. Optical Microscopy and SEM image of SiNWs dispersed in DI water and redeposited on silicon substrate.

- Define “porous and 1D feature”: how is the array of silicon nanowires 1D?

Response: The wires are used by dispersing them in DI water, as we can see from SEM and TEM images, these are porous nanowires. “1D” is used to describe the dimensionality of wires in the solution, not in the array format.

- Describe the “critical dimensions” of the porous nanowires: diameter, length, pore size, etc...

Response: The typical diameters of the wires is around ~100-200 nm, the length is on the order of 10 μm depending on the etching time, pore size is typically 5~10 nm from the high magnification TEM.

- Is the SiO₂ layer enough to stabilize SiNWs in solution? The authors claim that the SiO₂ layer is hydrophilic and there are pores on the wires (from the TEM in Figure 1). However, the TEM image is not very clear and should be magnified in order to demonstrate that the pores are on the wire and not on the silicon wafer surface.

Response: The TEM of SiNWs show that the wires are actually porous themselves amorphous surface oxide. The structure of such porous SiNWs are well characterized in previous studies (Nano letters 9 (12), 4539; ChemSusChem 5 (1), 177-180). These wires dispersed well in pure water solution without the help of surfactant, therefore we believe the wires are hydrophilic rendered by the surface oxide layer.

Figure R2-3. TEM images of SiNWs showing porous nature and thin oxide on surface under different magnification.

Comment 4: GNS:

- Please show TEM images (as Supp Inf) from the 3 stages of the preparation: (i) silica NP; (ii) silica NP with Au seeds; and (iii) silica NP with Au shell.

Response: We added below SEM and TEM images in the manuscript to illustrate the bare SiO₂ NP, SiO₂ NP with Au seeds and silica/Au nanoshell particles. From the SEM contrast, we can clear identify the insulating silica surface, Au seeds on the silica surface and the fully covered by rough Au surface.

Figure R2-4. SEM images of bare SiO₂ beads (a), with Au nanoseeds (b) and gold nanoshells (c). The inset in b shows the TEM image.

- What does relative rough surface mean? What is the difference between GNS and silica NP either by TEM image or other methods (AFM?). What is the thickness/roughness of the gold shell? Authors claim that silica nanospheres are 120 nm, but at the same time the gold nanoshells are also 120 nm.

Response: Thanks for the insightful point. Rough surface means that the Au nanoshell was made of multiple grains merged together, which may be initiated from different seeds, as we can see from Fig. 1b and Fig. R2-4. A careful analysis of the corresponding SEM images shows that the bare silica nanoparticle is about 114 nm in average, and the silica/Au core/shell nanoparticle is about 150 nm, with a shell thickness ~17nm. We are sorry for the typo and have corrected the relevant numbered in the manuscript accordingly.

Figure R2-5. Size distribution of GNS.

- *Modification of silica core with gold seeds: Is the solution kept in the dark? Given the color change, what is the size of the Au nanoparticles? UV vis measurement or DLS (dynamic light scattering) should provide a quick evidence. What is the composition of the GNS solution that is deposited on the LDI target plate?*

Response: The step of gold seeds synthesis was kept in dark for 24h. After that, the following synthesis steps don't require dark environment. The overall size of the GNS are about 150 nm, which consists 5-10 nm gold nanoparticles deposited from silica spheres. The UV-VIS absorption shows a shoulder peak around 500 nm, which is roughly consistent the plasmonic absorption band of ~4 nm Au nanoparticles. According to TEM, the size of the seeds are in the range of 2-10 nm in diameter (J. Phys. Chem. C 2016, 120, 377-385, Chem. Mater. 2016, 28, 1066-1075). The concentration of the GNS is about 0.25 mg/mL. We removed the unreacted chemicals from the GNS solution by centrifuge and wash for three times, and redispersed the GNS particles in DI water. Therefore, the GNS solution consists of GNS particles and water.

Figure R2-6. a, UV-Vis spectrum(a), TEM (b) and size distributions (c) of Au nanoseeds

Comment 5: - *Why the GNS ionization is different? Which part of the GNS mostly contributes to the LDI MS ionization efficiency? Is it only the gold? Can you omit the silica nanoparticle core?*

Response: We believe the the shell part primarily contributes to the LDI MI ionization. Silica core cannot be omitted because silicon nanoparticle core is used as the template to support and aid the gold shell formation. Because the silica core is completely enclosed in the Au shell, therefore, we can't remove the core without damaging the gold shell. The silica core here is used mostly for structural support. According to Erodan E. *et al.* (Nano Letters 3, 1411-1415), the silica core has minor effect on the plasmonic field of the GNS particles, in addition to the fact that silica itself has little absorption in the UV region, therefore, we believe the silica core does not contribute significantly to the energy absorption, nor the ionization.

Comment 6: *How critical is the architecture of the nanomaterial for the ionization process? How is the interaction analyte-nanomaterial? Half microL of serum sample from an individual was spotted on the target plate followed by 1 uL of nanomaterials. How do you control the concentration of nanomaterials in a 1 microL solution?*

Response: We thank the reviewer for carefully reading the manuscript. The shell structure of GNS enhanced local EM field and absorption coefficient, thus improving the ionization efficiency. In the case of SiNW, the mesoporous structure largely reduce the dimension of silicon

nanocrystals into a few nanometers owing to the quantum confinement, thus increasing the energy absorption in the UV regime used by LDI. In addition, the porous structure also enlarges the surface area of the nanowire, increasing the chance of interaction between nanowire and analyte. (Nature 399(6733), 243-246) We didn't conduct specific surface treatment or modification on the nanomaterial, therefore we didn't expect selective interaction between specific analyte and the nanomaterial surface such as antibody-antigen interaction. The concentration is controlled by original synthesis process. The concentration of as synthesized GNS solution can be estimated by the input concentration of silica beads and can be further evaluated by UV-Vis. Both of these nanomaterials dispersed very well in water.

***Comment 7:** This also relates to the quantitative capabilities of this approach in serum samples (Figure 1e). What is the reproducibility of SiNWs and GNS-based mass spectra? LDI MS approaches are not particularly quantitative nor reproducible. What is the quantitative dynamic range? Some features in figure 3 show intensity distributions close to 0k. Is there any signal-to-noise (S/N) ratio threshold applied to filter ions for the SVM model? Why is Figure 3 showing absolute intensity values instead of normalized values?*

Response: Thanks for the important question. We have replaced Figure 3 with normalized values. To ensure reproducibility, evaluation parameters were set so that only spectra containing at least one peak with a resolving power of greater than 300 and a signal-to-noise ratio of more than 30 in the m/z range of 100-500 were accumulated. The overall performance of the mass spectrometer was checked every day using a “home-made” standard and a stocked human serum standard prior to each run. (please see revised supplementary materials for more details)

Reply to Referee #3:

Comments: The authors present some impressive results using laser desorption of serum samples from nanomaterial surfaces for the diagnosis of a variety of different cancers. The sample size and independent validation at a second collection site provide some confidence that the methodology has promise for identify patterns of metabolite signals that could be useful in cancer diagnosis.

Response: We thank the referee for carefully reading the manuscript, and raising a number of excellent points about our studies. We welcome the opportunity to address these questions, and describe the changes we have made accordingly in the manuscript.

Comment 1: However, the fact that none of the signals were identified, nor even the total number of peaks used in the high dimensional analysis is a red flag. Also, the use of SVM, which generates an enormous number of higher order terms, none of which were specified in the analysis could cause problems. Given a mass spectrum containing hundreds of peaks, it's not hard to imagine that the classifier contains thousands of terms many of which are non-linear and which outnumber the number of samples. The problem is that when anything changes (and it will!) like the nano-material prep, the sample prep, the sample procurement protocol, the reaction products caused by laser irradiation of the sample, etc. it's easy for some of the non-linear terms to change dramatically, throwing off the algorithm.

Response: We thank the reviewer for carefully reading the manuscript and bringing up this important point. Inspired by the reviewer, we have carefully analyzed the m/z information of LDI-MS spectrum and identified 8 metabolite biomarkers with MS/MS, namely 2-oxovaleric acid, histamine, glucose, 5-hydroxymethyluracil, 2-Furoic acid, methylmalonic acid, methylcatechol and L-carnitine. Most of these metabolites were reported elsewhere for the link of cancer events (see revised manuscript for details). Some of the other m/z features may attribute to different ionization or fragments of the metabolites. For example, m/z = 219.08 could well be [glucose+K]⁺ and m/z = 105.00 could be fragment from histamine (Supplementary Fig. S9). This way, amongst all 120 m/z features, these 8 metabolites correspond up almost a third of the signals from Figure 3. It is also interesting to notice that some of the metabolites showed a preference for GNS or SiNW-assisted LDI. For example, 5-hydroxymethyluracil, which corresponded to m/z = 164.98, appeared 4 times in Figure 3 with GNS-assisted LDI while methylcatechol, which corresponded to m/z = 147.02, also appeared 4 times in Figures 3 with SiNW-assisted LDI, indicating different interaction mechanism between analytes and nanomaterials.

To ensure reproducibility, evaluation parameters were set so that only spectra containing at least one peak with a resolving power of greater than 300 and a signal-to-noise ratio of more than 30 in the m/z range of 100-500 were accumulated. The overall performance of the mass spectrometer was checked every day using a “home-made” standard and a stocked human serum standard prior to each run. (please see revised supplementary materials for more details)

Comment 2: It's also interesting to note that the authors don't try to use a classifier with a more limited number of peaks for the overall analysis, just heatmaps. Similarly, from the violin plots in Figure 3, there are very few strong biomarker candidates here, so that even if the authors were able to identify some of them, they would need many signals to make a suitable classifier.

The field of metabolomics has long moved on from the development of profiles based on a set of completely unknown peaks, because of the potential problems noted above, as well as the need for a biological connection between the metabolic changes observed and the disease state that can be verified or validated with further experiments.

For these reasons, I do not believe that the paper should be accepted for publication in Nature Communications. If, however, the authors are able to identify at least some of the major contributors to their model (i.e., the 30 metabolite signals they show in their heatmaps), then I think the paper would potentially be publishable, based on the positive qualities mentioned above.

Response: Thanks for the important question. These 8 metabolite signals contributes significantly in MNALCI, which represent up to ~30 signals from Figure 3. It is a very good point to try a classifier with more limited number of peak/signals, which will be further explored in our future work. As for the violin plots, several biomarkers were overlapped in different cancer types: for example, 4 times for 5-hydroxymethyluracil (m/z = 164.98) and 4 times for methylcatechol (m/z = 147.01). We set cut-off value of 3 for signal/noise ratio so that even if the relative intensity is weak, we still count that signal in as a valid signal.

Comment 3: Minor point: I don't think that the very high AUC values match with some of the sensitivity and specificity numbers provided. For example, it is not possible to generate an AUC of 0.99 using sensitivity=0.84 and specificity 0.84. The highest AUC could be in that situation is $1-(0.16)^2 = 0.97$, but it could definitely be lower depending on the shape of the AUC curve. The authors need to double check their other AUC values as well.

Response: Thanks for the opportunity to avoid any confusion. The calculation method of sensitivity 0.84 described in the manuscript is a little different from the usual method. The usual formula for calculating sensitivity is: $TP / (TP + FN)$, where the TP is the correct number of patients predicted by the model (True Positive) and FN is the wrong number of the healthy people predicted by the model (False Negative). The denominator represents the total number of patients. Since this is not a question of two categories, we need to identify specific diseases, so we only count those who are predicted to be positive and have the correct type of disease when we count TP in numerator. And the denominator are still the total number of patients. As a result, the calculated sensitivity is usually lower than that of binary classification.

Here is an example. There are 100 healthy control (HC) and 100 patients. There are four kinds of diseases (A/B/C/D), 25 people in each disease. Here is a confusion matrix of multi-classification:

	HC	A	B	C	D
HC	90	5	5	0	
A	5	20			
B		10	10	5	
C				25	
D				10	15

The sensitivity in our method is : $(20+10+25+15) / 100 = 0.7$

But if we regard it as a binary classification, the confusion matrix is:

	HC	Patients

HC	90	10
Patients	5	95

The sensitivity is: $95 / 100 = 0.95$

The sensitivity of 0.84 in our manuscript is the former, while the latter is used by the ROC and AUC.

Reviewers' Comments:

Reviewer #1:

Remarks to the Author:

This is a review report on the reviewed version of the manuscript entitled "Multiplexed Nanomaterial-Assisted Laser Desorption/Ionization for Pan-cancer Diagnosis and Classification", submitted for potential publication in the Nature Communications journal. The authors replied to all my comments, but some of them require further explanation.

Comment 2. To indicate that age, sex, and BMI do not influence the obtained classification model, the authors divided a healthy control group into two separate subgroups based on abovementioned parameters. As presented in Figure R1-1, MNALCI-based binary classification models of age, sex and BMI showed no significant difference between the two subgroups in any of these three comparisons. Was the model used for these calculations based on all signals or top selected signals? Was the classification of controls and cases by MNALCI in case of age, sex, and BMI-matched patients? Which signals are the top discriminators in case of groups selected in such a way? The other approach could be an adjustment of calculated p-values to abovementioned variables. Where the p-values presented in Table S8 corrected for multiple testing?

Comment 3. There is still no information about the calculation of the signal's reproducibility. As in the end, the intensities of signals are being compared; it is crucial to know how reproducible is this measurement for each metabolite. In the classical approach to metabolomics data, to perform statistical analysis, only reproducibly measured signals are selected.

Comment 4. These red dotted lines present on the fragmentation spectra hide some of the signals. Fragmentation spectra of metabolite present in plasma and that of reference compound should be overlapping. Please add the signal size to the Y-axis. In the case of serum samples, the signals originating from fragmented ion can be selected by use of correlation. All fragmentation signals are highly correlated with a parent ion. Even that identification of some of the signals was not possible, fragmentation of all top signals should be provided.

Reviewer #2:

Remarks to the Author:

Most issues have been properly addressed, particularly those referring to the nanostructured matrices. Perhaps the characterization of the nanomaterials could be better explained in the supplementary material, putting together the responses of the rebuttal letter. This would facilitate the understanding of the new figures and results.

However, there's a critical point that remains to be solved: it is now reported that 8 metabolites were identified and confirmed by MS/MS, namely 2-oxovaleric acid, histamine, glucose, 5-hydroxymethyluracil, 2-Furoic acid, methylmalonic acid, methylcatechol and L-carnitine. Unfortunately, these experiments were conducted using TOF/TOF LIFT, which is a very low quality technique for MS/MS. Not only the spectra are horrible and do not meet any minimum standard of quality for metabolite identification, but also some of the reported metabolites are very suspicious of being endogenous compounds with the potential to distinguish different types of cancers. For instance, 2-Furoic acid is an organic compound most widely found in food products as a preservative and a flavoring agent. There are two main isomers of methylcatechol (3-methyl and 4-methylcatechol) and both are exogenously produced. 4-methylcatechol is present in a few different foods, and 3-methylcatechol is a xenobiotic metabolite produced by some bacteria capable of degrading nitroaromatic compounds present in pesticide-contaminated soil samples. The authors justify the non-use of LC-MS/MS because "the quantification of analytes using LC-MS might be altered by pretreatment procedure while LDI MS/MS remained identical with the protocol of the samples". Whereas I can partially agree with this response, this does not invalidate the use of LC-MS/MS (e.g., LC-QqQ MS using MRM with standards) to confirm the presence and unequivocal identification of the 8 metabolites. Clearly, glucose, L-carnitine and histamine are well described metabolites in human serum, but some of the others are not so well reported, or not reported at all.

Unfortunately, until the identity of the metabolite biomarkers are correctly demonstrated I cannot

support the publication of this paper.

Reviewer #3:

Remarks to the Author:

I appreciate that the authors have now identified 8 of the metabolites they are detecting and that their MS peaks and adducts correspond to a sizeable number of the 120 peaks used in the analysis. While some of these are quite interesting, others, like glucose, histamine, are known to be relatively weak cancer markers. As a result, and based on the violin plots of Figure 3, there is likely something that this model is doing that is not understood or explained by the authors to give the surprising performance. In any event, I have a few additional questions that need to be addressed prior to publication.

1) It doesn't appear that the authors did any of the standard evaluations of the instrument data quality. For example, what is the linearity, limit of detection and dynamic range of their detected metabolites. What is the reproducibility of the determined values, i.e., relative quantitation over a day or across multiple days? These are important to make sure that instrument bias is not contributing to the model.

2) Why are the MS/MS spectra in Figure S9 so complicated, i.e., with so much chemical noise? Are the standard samples, without the nanomaterials so complicated, or do the nanomaterials cause these spectra to be complicated?

3) Thank you for providing an explanation of the sensitivity and ROC results. I think it would be useful to tell the readers that the ROC curves are for a binary comparison as you do in the response to reviewers.

4) Please also label the known metabolites/adducts in the violin plots of Figure 3.

Reply to Referee #1:

Comments: This is a review report on the reviewed version of the manuscript entitled “Multiplexed Nanomaterial-Assisted Laser Desorption/Ionization for Pan-cancer Diagnosis and Classification”, submitted for potential publication in the Nature Communications journal. The authors replied to all my comments, but some of them require further explanation.

Response: We thank the referee for carefully reading the manuscript, and raising a number of excellent points about our studies. We welcome the opportunity to address these questions, and describe the changes we have made accordingly in the manuscript.

Comment 1: To indicate that age, sex, and BMI do not influence the obtained classification model, the authors divided a healthy control group into two separate subgroups based on abovementioned parameters. As presented in Figure R1-1, MNALCI-based binary classification models of age, sex and BMI showed no significant difference between the two subgroups in any of these three comparisons. Was the model used for these calculations based on all signals or top selected signals? Was the classification of controls and cases by MNALCI in case of age, sex, and BMI-matched patients? Which signals are the top discriminators in case of groups selected in such a way? The other approach could be an adjustment of calculated p -values to abovementioned variables. Where the p -values presented in Table S8 corrected for multiple testing?

Response: Thanks for bringing up this very important point. This set of figures were made for the last comment whether our MNALCI approach was highly affected by age, sex or BMI. The X axis used age, sex and BMI for variations, while Y axis represents MNAICI score. The p values by t-test showed no difference for MNALCI score, indicating weak influence by these factors ($p = 0.492$ for age, $p = 0.641$ for sex and $p = 0.367$ for BMI, respectively. Figure RR1-1) The model used all signals from the sample and all the patients with a BMI data was included in this classification. Since no significant difference was observed between any two subgroups, no top discriminator was calculated in this case.

Figure RR1-1. a, Binary classification models of age (a), sex (b) and BMI where Y axis represents MNALCI output. The probability shows no significant difference between two groups in any of these three figures.

Comment 2: There is still no information about the calculation of the signal’s reproducibility. As in the end, the intensities of signals are being compared; it is crucial to know how reproducible is this measurement for each metabolite. In the classical approach to metabolomics data, to perform statistical analysis, only reproducibly measured signals are selected.

Response: Thanks for the very insightful question. We have elaborated this part in the supplementary materials. Generally, the overall performance of the mass spectrometer was checked

every 9 samples as a group using a stocked human serum standard (aliquoted and frozen). In addition, a “home-made” standard consisting serine ($m/z=105.09$), glucose ($m/z=180.16$), tryptophan ($m/z=204.23$), sucrose ($m/z=342.29$), maltotriose ($m/z=504.44$) and amylopentaose ($m/z=828.72$) was test each run for calibration. After calibration for each run, all the standard spots were collected for a Principal Component Analysis (PCA) analysis, the results were considered consistent when the first principal components are within $\pm 2SD$ (Standard Deviation). Any outlier standard will need to be retested altogether with the other 8 samples in the same group.

***Comment 3:** These red dotted lines present on the fragmentation spectra hide some of the signals. Fragmentation spectra of metabolite present in plasma and that of reference compound should be overlapping. Please add the signal size to the Y-axis. In the case of serum samples, the signals originating from fragmented ion can be selected by use of correlation. All fragmentation signals are highly correlated with a parent ion. Even that identification of some of the signals was not possible, fragmentation of all top signals should be provided.*

Response: Thanks, and we agree these are important details and have now modified in the revised supplementary materials.

Reply to Referee #2:

Comments: Most issues have been properly addressed, particularly those referring to the nanostructured matrices.

Response: We thank the referee for carefully reading the manuscript, and raising a number of excellent points about our studies. We welcome the opportunity to address these questions, and describe the changes we have made accordingly in the manuscript.

Comment 1: Perhaps the characterization of the nanomaterials could be better explained in the supplementary material, putting together the responses of the rebuttal letter. This would facilitate the understanding of the new figures and results.

Response: Thanks, and we agree these are important details. We have now added this part in the manuscript, the figure caption and the method section of the supplementary materials.

Comment 2: However, there's a critical point that remains to be solved: it is now reported that 8 metabolites were identified and confirmed by MS/MS, namely 2-oxovaleric acid, histamine, glucose, 5-hydroxymethyluracil, 2-Furoic acid, methylmalonic acid, methylcatechol and L-carnitine. Unfortunately, these experiments were conducted using TOF/TOF LIFT, which is a very low-quality technique for MS/MS. Not only the spectra are horrible and do not meet any minimum standard of quality for metabolite identification, but also some of the reported metabolites are very suspicious of being endogenous compounds with the potential to distinguish different types of cancers. For instance, 2-Furoic acid is an organic compound most widely found in food products as a preservative and a flavoring agent. There are two main isomers of methylcatechol (3-methyl and 4-methylcatechol) and both are exogenously produced. 4-methylcatechol is present in a few different foods, and 3-methylcatechol is a xenobiotic metabolite produced by some bacteria capable of degrading nitroaromatic compounds present in pesticide-contaminated soil samples. The authors justify the non-use of LC-MS/MS because "the quantification of analytes using LC-MS might be altered by pretreatment procedure while LDI MS/MS remained identical with the protocol of the samples". Whereas I can partially agree with this response, this does not invalidate the use of LC-MS/MS (e.g., LC-QqQ MS using MRM with standards) to confirm the **presence and unequivocal identification** of the 8 metabolites. Clearly, glucose, L-carnitine and histamine are well described metabolites in human serum, but some of the others are not so well reported, or not reported at all. Unfortunately, until the identity of the metabolite biomarkers are correctly demonstrated I cannot support the publication of this paper.

Response: We appreciate the very insightful comment. Many literatures reported the application of MALDI-TOF/TOF-MS in biological metabolites identification, including small molecules involved in phosphorylation, bacterial proteins, and cocaine in hair, *etc.* ⁽¹⁻⁸⁾. Inspired by the reviewer request, we have also conducted LC-MS studies on the other five metabolites and included these data in the supplementary materials (Fig. S16) to further validate the presence of these metabolites, including addition to the three metabolites mentioned by the reviewer, 2-Furoic acid, 2-oxovaleric, methylmalonic acid, 4-Methylcatechol, and 5-hydroxymethyluracil. The verification of these metabolites was further achieved by comparing the m/z features with human metabolome database ([HMDB0000617](https://pubchem.ncbi.nlm.nih.gov/compound/2-Furoic-acid) for 2-Furoic acid, [HMDB0001865](https://pubchem.ncbi.nlm.nih.gov/compound/2-Oxovaleric-acid) for 2-oxovaleric, [HMDB0000202](https://pubchem.ncbi.nlm.nih.gov/compound/Methylmalonic-acid) for methylmalonic acid, [HMDB0000873](https://pubchem.ncbi.nlm.nih.gov/compound/4-Methylcatechol) for 4-Methylcatechol, and [HMDB0000469](https://pubchem.ncbi.nlm.nih.gov/compound/5-Hydroxymethyluracil) for 5-hydroxymethyluracil, respectively). We agree with the reviewer that these metabolites may not be all endogenous compounds, but these are in fact metabolites existed in the human blood and proved to

be correlated to our MNALCI algorithm in this study. We believe and it has also been previously suggested that not only endogenous compounds have the potential to distinguish cancer types, but exogenous ingredients also play an important role for disease occurrence and development, reflect human metabolism level in some degree and could be critical diagnostic markers. For example, endocrine disrupting chemical Bisphenol A is an exogenous compound but can be detected in human blood. And it is known that Bisphenol A has been associated with serious health effects in humans and wildlife: it elicits several endocrine disorders and affects the pathogenesis of several hormone-dependent tumors such as breast, ovarian, prostate cancer and others.⁽⁹⁾

Together, we agree that there are more advanced analytic techniques out there. It should be noted that the very capability to achieve high fidelity cancer identification from highly complex spectra with many interfering species is impressive and demonstrates the power of our approach. The applications of our parametric analysis approach with other advanced analytic techniques could further expand the capability of this approach and will be an important topic in the future discovery.

References:

1. Chong YK, Ho CC, Leung SY, Lau SKP, Woo PCY. Clinical Mass Spectrometry in the Bioinformatics Era: A Hitchhiker's Guide. *Comput Struct Biotechnol J*. 2018;16:316-334.
2. Leonardi A, Palmigiano A, Mazzola EA, Messina A, Milazzo EM, Bortolotti M, Garozzo D. Identification of human tear fluid biomarkers in vernal keratoconjunctivitis using iTRAQ quantitative proteomics. *Allergy*. 2014;69(2):254-260.
3. Smith AM, Awuah E, Capretta A, Brennan JD. A matrix-assisted laser desorption/ionization tandem mass spectrometry method for direct screening of small molecule mixtures against an aminoglycoside kinase. *Anal Chim Acta*. 2013;786:103-110.
4. Fagerquist CK, Garbus BR, Miller WG, Williams KE, Yee E, Bates AH, Boyle S, Harden LA, Cooley MB, Mandrell RE. Rapid identification of protein biomarkers of Escherichia coli O157:H7 by matrix-assisted laser desorption ionization-time-of-flight-time-of-flight mass spectrometry and top-down proteomics. *Anal Chem*. 2010;82(7):2717-2725.
5. Kernalleguen A, Steinhoff R, Bachler S, Dittrich PS, Saint-Marcoux F, El Bakhi S, Vorspan F, Leonetti G, Lafitte D, Pelissier-Alicot AL, Zenobi R. High-Throughput Monitoring of Cocaine and Its Metabolites in Hair Using Microarrays for Mass Spectrometry and Matrix-Assisted Laser Desorption/Ionization-Tandem Mass Spectrometry. *Anal Chem*. 2018;90(3):2302-2309.
6. Fresnais M, Roth A, Foerster KI, Jager D, Pfister SM, Haefeli WE, Burhenne J, Longuespee R. Rapid and Sensitive Quantification of Osimertinib in Human Plasma Using a Fully Validated MALDI-IM-MS/MS Assay. *Cancers (Basel)*. 2020;12(7).
7. Huang L, Gurav DD, Wu S, Xu W, Vedarethinam V, Yang J. A Multifunctional Platinum Nanoreactor for Point-of-Care Metabolic Analysis. *Matter*. 2019;1(6):1669-1680.
8. Huang L, Wang L, Hu X, Chen S, Tao Y, Su H, Yang J, Xu W, Vedarethinam V, Wu S, Liu B, Wan X, Lou J, Wang Q, Qian K. Machine learning of serum metabolic patterns encodes early-stage lung adenocarcinoma. *Nat Commun*. 2020;11(1):3556.
9. Shafei A, Ramzy MM, Hegazy AI, Husseny AK, El-Hadary UG, Taha MM, Mosa AA. The molecular mechanisms of action of the endocrine disrupting chemical bisphenol A in the development of cancer. *Gene*. 2018;20(647):235-243.

Reply to Referee #3:

Comments: I appreciate that the authors have now identified 8 of the metabolites they are detecting and that their MS peaks and adducts correspond to a sizeable number of the 120 peaks used in the analysis. While some of these are quite interesting, others, like glucose, histamine, are known to be relatively weak cancer markers. As a result, and based on the violin plots of Figure 3, there is likely something that this model is doing that is not understood or explained by the authors to give the surprising performance. In any event, I have a few additional questions that need to be addressed prior to publication.

Response: We thank the referee for carefully reading the manuscript, and raising a number of excellent points about our studies. We welcome the opportunity to address these questions, and describe the changes we have made accordingly in the manuscript.

Comment 1: It doesn't appear that the authors did any of the standard evaluations of the instrument data quality. For example, what is the linearity, limit of detection and dynamic range of their detected metabolites. What is the reproducibility of the determined values, i.e., relative quantitation over a day or across multiple days? These are important to make sure that instrument bias is not contributing to the model.

Response: Thanks for the very important question. We have further elaborated this part in the supplementary materials. Generally, the overall performance of the mass spectrometer was checked every 9 samples as a group using a stocked human serum standard (aliquoted and frozen). In addition, a "home-made" standard consisting serine ($m/z=105.09$), glucose ($m/z=180.16$), tryptophan ($m/z=204.23$), sucrose ($m/z=342.29$), maltotriose ($m/z=504.44$) and amylopentaose ($m/z=828.72$) was test each run for calibration. After calibration for each run, all the standard spots were collected for a Principal Component Analysis (PCA) analysis, (both for the same run and between different days) the results were considered consistent when the first principal components are within $\pm 2SD$ (Standard Deviation). Any outlier standard will need to be retested altogether with the other 8 samples in the same group.

Comment 2: Why are the MS/MS spectra in Figure S9 so complicated, i.e., with so much chemical noise? Are the standard samples, without the nanomaterials so complicated, or do the nanomaterials cause these spectra to be complicated?

Response: We appreciate the insightful comment. For the secondary MS/MS spectra, we believe these complicated spectra figures may be due to the following reasons⁽¹⁻⁹⁾: (1) The signal intensity of the target peak obtained was relatively low, which lead to relatively high background noise; (2) The low content of the target substance in the serum leads to the weak signal obtained. The mixture of the sample and matrix contains many other substances of the similar molecular weight, which leads to the complex secondary mass spectrometry. (3) Similar phenomenon exists in the secondary mass spectrometry of many newly developed matrices applied to MALDI, and the exact mechanism needs to be further studied.. The standard samples need matrix materials to generate any signal, and therefore we cannot obtain MS/MS spectra without matrix. Even the traditionally widely used matrix materials (CHCA) generated little signal (Fig. 1e purple line). We believe the noise and complicated MS/MS spectra were largely attributed to the complexity of the serum sample and low signal intensity of the tarted species of interest. **The very capability to achieve high fidelity cancer identification from highly complex spectra with many interfering species is impressive and demonstrates the power of our approach.**

Comment 3: Thank you for providing an explanation of the sensitivity and ROC results. I think it

would be useful to tell the readers that the ROC curves are for a binary comparison as you do in the response to reviewers.

Response: Thanks for bringing up this very important point. We have now added this part in the manuscript.

Comment 4: Please also label the known metabolites/adducts in the violin plots of Figure 3.

Response: Thanks, and we agree these are important details and have modified in the revised manuscript.

References:

1. Lu M, Yang X, Yang Y, Qin P, Wu X, Cai Z. Nanomaterials as Assisted Matrix of Laser Desorption/Ionization Time-of-Flight Mass Spectrometry for the Analysis of Small Molecules. *Nanomaterials (Basel)*. 2017;7(4).
2. Huang L, Gurav DD, Wu S, Xu W, Vedarethinam V, Yang J. A Multifunctional Platinum Nanoreactor for Point-of-Care Metabolic Analysis. *Matter*. 2019;1(6):1669-1680.
3. Huang L, Wang L, Hu X, Chen S, Tao Y, Su H, Yang J, Xu W, Vedarethinam V, Wu S, Liu B, Wan X, Lou J, Wang Q, Qian K. Machine learning of serum metabolic patterns encodes early-stage lung adenocarcinoma. *Nat Commun*. 2020;11(1):3556.
4. Chong YK, Ho CC, Leung SY, Lau SKP, Woo PCY. Clinical Mass Spectrometry in the Bioinformatics Era: A Hitchhiker's Guide. *Comput Struct Biotechnol J*. 2018;16:316-334.
5. Leonardi A, Palmigiano A, Mazzola EA, Messina A, Milazzo EM, Bortolotti M, Garozzo D. Identification of human tear fluid biomarkers in vernal keratoconjunctivitis using iTRAQ quantitative proteomics. *Allergy*. 2014;69(2):254-260.
6. Smith AM, Awuah E, Capretta A, Brennan JD. A matrix-assisted laser desorption/ionization tandem mass spectrometry method for direct screening of small molecule mixtures against an aminoglycoside kinase. *Anal Chim Acta*. 2013;786:103-110.
7. Fagerquist CK, Garbus BR, Miller WG, Williams KE, Yee E, Bates AH, Boyle S, Harden LA, Cooley MB, Mandrell RE. Rapid identification of protein biomarkers of Escherichia coli O157:H7 by matrix-assisted laser desorption ionization-time-of-flight-time-of-flight mass spectrometry and top-down proteomics. *Anal Chem*. 2010;82(7):2717-2725.
8. Kernalleguen A, Steinhoff R, Bachler S, Dittrich PS, Saint-Marcoux F, El Bakhi S, Vorspan F, Leonetti G, Lafitte D, Pelissier-Alicot AL, Zenobi R. High-Throughput Monitoring of Cocaine and Its Metabolites in Hair Using Microarrays for Mass Spectrometry and Matrix-Assisted Laser Desorption/Ionization-Tandem Mass Spectrometry. *Anal Chem*. 2018;90(3):2302-2309.
9. Fresnais M, Roth A, Foerster KI, Jager D, Pfister SM, Haefeli WE, Burhenne J, Longuespee R. Rapid and Sensitive Quantification of Osimertinib in Human Plasma Using a Fully Validated MALDI-IM-MS/MS Assay. *Cancers (Basel)*. 2020;12(7).

Reviewers' Comments:

Reviewer #1:

Remarks to the Author:

I have evaluated the revised version of the manuscript together with the answers of the authors. While I am satisfied with the answers to comments 2 and 3, I still insist to include in the supplementary materials file the MS/MS fragmentation spectra of at least each of the Top 10 discriminatory features presented in Figure 3. Although these m/z values remain unidentified for the authors, other researchers may be able to indicate, based on the characteristic fragmentation pattern, at least a class of metabolites to which they belong. As they are crucial signals, the maximal possible information about them should be presented.

Reviewer #2:

Remarks to the Author:

It is incomprehensible that the authors have used a QTOF mass spectrometer (Bruker Impact II) with MS/MS capabilities, and they have not used these technical capabilities to validate the identity of metabolites. Not only that, they claim "LC-MS measurements" but in reality there is no LC separation. It's all direct infusion in full scan mode.

The experiment that needs to be done is very simple:

- 1) Run the standards on a LC-MS system and keep the retention time (RT) values for each compound.
- 2) Run MS/MS experiments on the standards using your QTOF MS.
- 3) Run the serum extract under identical conditions and verify that both RT and MS/MS pattern are the same for each metabolite.

If this is done and properly shown, I'm in favor of publication.

Reviewer #3:

Remarks to the Author:

The authors have answered my questions satisfactorily

plots of Figure 3.

Reply to Referee #1:

Comments: I still insist to include in the supplementary materials file the MS/MS fragmentation spectra of at least each of the Top 10 discriminatory features presented in Figure 3. Although these m/z values remain unidentified for the authors, other researchers may be able to indicate, based on the characteristic fragmentation pattern, at least a class of metabolites to which they belong. As they are crucial signals, the maximal possible information about them should be presented.

Response: Thanks for bringing up this very important point. We have now included all MS/MS spectra of the top 10 discriminatory features (Fig S24-S35).

Reply to Referee #2:

Comment: *It is incomprehensible that the authors have used a QTOF mass spectrometer (Bruker Impact II) with MS/MS capabilities, and they have not used these technical capabilities to validate the identity of metabolites. Not only that, they claim "LC-MS measurements" but in reality there is no LC separation. It's all direct infusion in full scan mode.*

The experiment that needs to be done is very simple:

- 1) Run the standards on a LC-MS system and keep the retention time (RT) values for each compound.*
- 2) Run MS/MS experiments on the standards using your QTOF MS.*
- 3) Run the serum extract under identical conditions and verify that both RT and MS/MS pattern are the same for each metabolite.*

If this is done and properly shown, I'm in favor of publication.

Response: Thanks, and we agree these are important details. We have now added retention time and LC MS/MS spectra for all eight metabolites identified in the manuscript (Fig S16-S23). The results showed that both RT and MS/MS patterns in serum samples matched the data for those in the standards under identical conditions.

Reviewers' Comments:

Reviewer #2:

Remarks to the Author:

I'm in favor of publication